# A modified modeling and dynamical behavior analysis method for fractional-order positive Luo converter

Zirui Jia [ID]*, Chongxin Liu

State Key Laboratory of Electrical Insulation and Power Equipment, Xi'an Jiaotong University, Xi'an, Shannxi, China

* jiazirui@126.com

## Abstract

Compared to the integer-order modeling, the fractional-order modeling can achieve higher accuracy for designing and analyzing the DC-DC power converters. However, its applications in pulse width modulation (PWM) converters are limited due to the computational complexities. In this paper, a modified fractional-order modeling methodology for DC-DC converters is proposed, and its effectiveness is verified on the fractional-order positive Luo converters. Instead of using fractional-order calculus, the proposed methodology analyzes the harmonic components of the PWM converters by utilizing the non-linear vector differential equations of the periodically time-variant system. The final solution of the state variables is composed of two parts: the steady-state solution and the transient solution. The approximate steady state solution can be obtained by using the equivalent small parameter (ESP) method and the harmonic balance theory, while the main part of the transient solution can be obtained according to the explicit Grünwald-Letnikov (GL) approximation. In addition, the influence of the fractional orders on the performance of the DC-DC converters, and on the dynamic behaviors of the fractional-orders systems are also discussed in this paper. Compared to the conventional fractional-order numerical models, the proposed model is able to present the time-domain information more precisely, which helps to better reveal and analyze the non-linear behaviors of the DC-DC converters. The effectiveness of the work is demonstrated by the simulation and experimental results of the equivalent circuits built with fractional-order components.

## Introduction

In recent years, the findings, of the fractional-order inductive phenomena, in physics, engineering, biology and other fields, have led to a closer relationship between theory and practice [1–4]. Numerous mathematical modeling studies on the passive components in [5, 6], i.e. inductors and capacitors, have shown that, compared to the integer-order modeling, the fractional order modeling methodology can better present the electrical characteristics of the system, making it a widespread research topic [7]. In addition, since inductors and capacitors

**Data Availability Statement:** All relevant data are within the paper and its Supporting Information files.

**Funding:** This work was supported in part by the National Nature Science Foundation of China under

Grant No. 51877162, http://www.nsfc.gov.cn. The funders had no role in study design, data collection and analysis, decision to publish, or preparation of the manuscript. No additional external funding received for this study.

play important roles in the power converters, it is more accurate to model the power converters with the fractional-order modeling method [8–16]. So far, there are only a few commercially available fractional-order passive components. The fractional-order modeling for passive components are developed from empirical results [17, 18], or equivalent models of the passive components described by the fractional-order definition [19–24].

Fractional-order models for different DC-DC converters in different modes given in [8–13] are using fractional calculus and state space averaging techniques. By utilizing the fractional-order definitions, the mean and ripple values of the steady-state variables can be obtained, but the transient solutions cannot be derived. Because of the inconsistent definitions of the fractional-order derivative, such modeling technique for the fractional-order converters is not always valid. Besides, [12, 13] discussed the modified Oustaloup's approximation in fractional integral module, which is used to acquire the transient solutions. Without discretizing, this method is a precise engineering simulation for the transient responses analysis, but is not an appropriate solution for the non-linear behaviors analysis due to the frequency domain approximation. [14] offers a fractional means to characterize the non-solid aluminum electrolytic capacitors in DC-DC converters. On the other hand, the Predictor-Corrector Adams-Bashforth-Moulton (PECE-ABM) type numerical method is often used to obtain the solutions of the fractional systems. Due to the "long memory" characteristics of the fractional-order modeling, the derivation of the fractional-order calculus is usually not straightforward. The approach to get the steady-state variables requires processing the whole datapoints between the initial state and the steady state. Therefore, large amounts of computation efforts are involved in the solving process. The time-domain modeling proposed in [15, 16] uses the simplified equivalent small parameter (SESP) method to get the steady state waveforms, instead of using the fractional-order derivative definitions. It is able to solve the steady state variables without circuit simulations or multiple iterations. However, its accuracy still needs improvement and it does not apply to the situation in which the system state variable changes abruptly in one switching cycle. In addition, the transient solutions of the fractional model are not mentioned.

[12], [13] and [25] studied the dynamic behaviors of the fractional-order PWM converters, but their results are obtained via the *MATLAB/Simulink* simulations using the approximation of the fractional-order components. Furthermore, none of the above-mentioned literatures have investigated the non-linear behaviors via numerical analysis in time-domain. Ref. [26] presented extensive experimental results, which exposes the operating mechanism of the limit cycle behavior in the integer-order boost converters. Nevertheless, its conclusions do not apply to the Grünwald-Letnikov (a.k.a. Riemann-Liouville and Caputo) definition-based systems. These fractional-order systems do not have periodic solutions [27–30], but only have the asymptotically limit cycle behaviors. Considering their impact on the device stress and the system efficiency during the switching periods, such asymptotically limit cycle behaviors require special attention in practical applications.

This paper is focusing on the asymptotically limit cycle behaviors in the fractional-order converters with the most widely used proportional-integral (PI) voltage compensator. Taking the positive Luo converter as an example, a modified time-domain modeling and analyzing methodology for DC-DC converters is proposed. The theoretical foundation of the work presented here is based on Eq (1) from [31],

$$\frac{\mathrm{d}^{\lambda} e^{\omega t}}{\mathrm{d}t^{\lambda}} = \omega^{\lambda} e^{\omega t} \tag{1}$$

in which, $\lambda$ is the order, and $\omega$ is the angular frequency of above differential operator. Noticeably, $\lambda$ can be either an integer or a fraction.

Eq (1) indicates that the differential operator only influences the amplitude, but not the angular frequency in the exponential function. Because of this, the equivalent small parameter (ESP) method and the harmonic balance theory can be applied to the fractional differential operation. By extending the conventional ESP method into the fractional domain, a modeling method and the steady state solutions of the fractional-order DC-DC converters can be obtained. Therefore, the final approximate analytical solutions of the PWM converters in the steady state can be represented as the sum of the harmonic contents. Different from basic DC-DC converters, the voltage across the transfer capacitors in the positive Luo converter exhibits huge oscillations when the power switch turns off. The filtering property of the positive Luo converter is not as strong as that in other PWM converters. Since the first three terms, $x_0$, $x_1$ and $x_2$, in [32], Eq (4) cannot ensure the accuracy of the state variables in positive luo converters, a modified algorithm by extending the consideration scope to more equations of the equivalent system is proposed in this paper. Compared with conventional schemes, the proposed method is able to solve the steady state variables of the positive Luo converter more accurately. With the explicit Grünwald-Letnikov (GL) approximation, the proposed method greatly reduces the number of iterations when computing the transient solutions. Therefore, the speed and the accuracy of the computations can be improved at the same time. Since the proposed model is developed in the time domain, and the correction coefficients are used in the case of a homogeneous initial value, it can uncover the nonlinear fractional-order behaviors more comprehensively and realistically.

This paper is organized as follows: Section 2 introduces the mathematical modeling method for the fractional-order positive Luo converters operated in CCM, and their equivalent circuit models based on the ESP method. Section 3 presents the procedure to get the steady state and transient solutions of the fractional-order positive Luo converter. Section 4 provides the detailed numerical simulation results, which demonstrate the effectiveness of the proposed method. Since the fractional orders have great influence on the CCM-operating criterion, the transfer functions and the harmonic amplitudes of the state variables of the converters, these order-related phenomena are discussed in detail in Section 5. The asymptotically limit cycle behavior analysis of the PI controlled fractional-order positive Luo converter is shown in section 6. Section 7 carries out circuit simulations and experiments which further demonstrate the above-mentioned analysis. Section 8 summarizes the conclusion and future work.

## Equivalent model of the fractional-order positive Luo converter using the ESP method

According to the linear capacitor model and the inductor model proposed by Westerlund [33, 34], all real capacitors and inductors are fractional essentially. The voltage across a real inductor $v_L$ and the current through a real capacitor $i_C$ can be, respectively, described by

$$v_L = L\frac{\mathrm{d}^\lambda i_L}{\mathrm{d}t^\lambda}, 0 < \lambda < 1 \tag{2a}$$

$$i_C = C\frac{\mathrm{d}^\mu v_C}{\mathrm{d}t^\mu}, 0 < \mu < 1 \tag{2b}$$

where $L$ presents the inductor's inductance, $C$ is the capacitor's capacitance, and $\lambda$, $\mu$ are the orders. They have relationships with the "proximity effect" and the kind of the dielectric.

As shown in Fig 1, circuit for the positive output elementary super lift Luo converter consists of the DC supply voltage $V_{in}$, the inductor $L$, the capacitors $C_b$ and $C_o$, the power switch $S$,

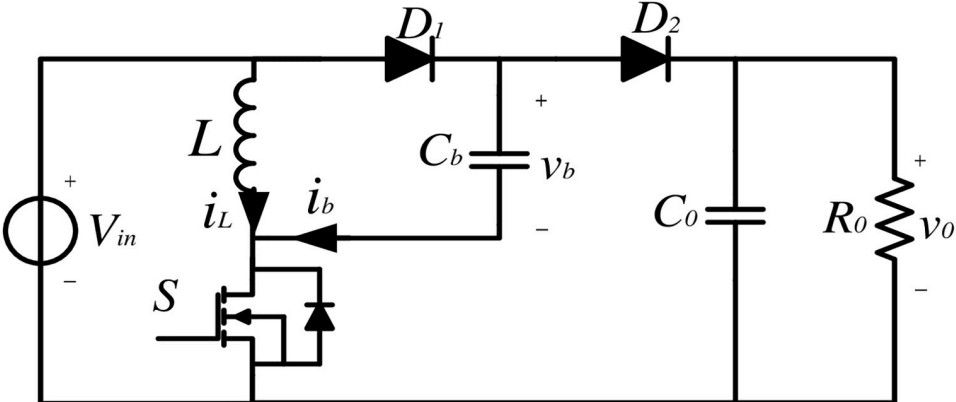

**Fig 1. Circuit of the positive Luo converter.**

the diodes $D_1$ and $D_2$, and the load resistor $R_o$. During the analysis of the operating process, all the components are assumed to be ideal and running in CCM mode.

Let $n$ be any integer, $T$ represent the switching period and $D$ denote the duty ratio in the steady state. Generally speaking, there are two switching states for the positive Luo converter in CCM mode, identified as:

State 1: $t \in (nT, (n + D)T]$, during which $S$ and $D_1$ are on, and $D_2$ is reversed-biased; the equivalent circuit is shown in Fig 2(a).

State 2: $t \in ((n + D)T, (n + 1)T]$, during which $S$ and $D_1$ are off, and $D_2$ is forward-biased; the equivalent circuit is presented in Fig 2(b).

In State 1, during the on-time period of the switch $S$, capacitor $C_b$ is charged by the voltage supply, and the current $i_L$ flowing through $L$ increases. In State 2, during the off-time interval of the switch, the inductor $L_1$ is still conducting current, and the capacitor $C_b$ and $C_o$ are discharged. It is worth mentioning that $R_{in}$ is the internal resistance of the voltage supply, which is a very small value but cannot be neglected in the model of this paper [35, 36]. To represent the switching state of the PWM converter, a periodic scalar function $s(t)$ is introduced, which

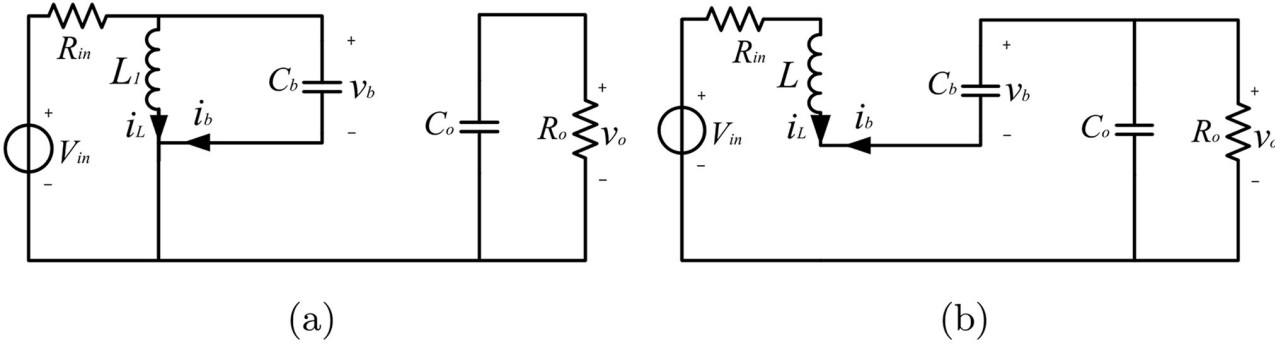

(a)                                                            (b)

**Fig 2. Operating states of the positive Luo converter: (a) State 1; (b) State 2.**

is denoted as

$$s(t) = \begin{cases} 0 & t \in (nT, (n+D)T] \\ \\ 1 & t \in ((n+D)T, (n+1)T] \end{cases} \tag{3}$$

Select the inductor current $i_L$, the capacitor voltages $v_b$ and $v_o$ as the state variables. In State 1, the state equations of the positive Luo converter can be described as

$$\frac{d^\alpha i_L}{dt^\alpha} - \frac{1}{L} v_b = 0 \tag{4a}$$

$$\frac{d^\beta v_o}{dt^\beta} + \frac{1}{R_o C_o} v_o = 0 \tag{4b}$$

$$\frac{d^\gamma v_b}{dt^\gamma} + \frac{1}{C_b} i_L + \frac{1}{R_{in} C_b} v_b = \frac{1}{R_{in} C_b} V_{in} \tag{4c}$$

When operating in State 2, the state equations are expressed as

$$\frac{d^\alpha i_L}{dt^\alpha} + \frac{R_{in}}{L} i_L + \frac{1}{L} v_o - \frac{1}{L} v_b = \frac{1}{L} V_{in} \tag{5a}$$

$$\frac{d^\beta v_o}{dt^\beta} - \frac{1}{C_o} i_L + \frac{1}{R_o C_o} v_o = 0 \tag{5b}$$

$$\frac{d^\gamma v_b}{dt^\gamma} + \frac{1}{C_b} i_L = 0 \tag{5c}$$

In order to simplify the equations, we use $p$ to replace $d/dt$. Correspondingly, $d^\alpha/dt^\alpha$, $d^\beta/dt^\beta$ and $d^\gamma/dt^\gamma$ are represented by $p^\alpha$, $p^\beta$ and $p^\gamma$, respectively. In this way, the converter in the steady state can be formulated by the following vector differential equation of the state variable

$$\mathbf{G}_1(p^\alpha, p^\beta, p^\gamma)\mathbf{x} + \mathbf{G}_2 \mathbf{f}(\mathbf{x}) = \mathbf{H}_1 + s(t)\mathbf{H}_2 \tag{6}$$

Here, the vector of the state variables is $\mathbf{x} = [i_L \ v_o \ v_b]^T$, and the non-linear vector function is indicated as $\mathbf{f}(\mathbf{x}) = s(t)\mathbf{x}$. $\mathbf{H}_1 = [V_{in}/L \ 0 \ 0]^T$ and $\mathbf{H}_2 = [-V_{in}/L \ 0 \ V_{in}/(R_{in} C_b)]^T$ are the constant vectors. The matrices $\mathbf{G}_1(p^\alpha, p^\beta, p^\gamma)$ and $\mathbf{G}_2$ are denoted by

$$\mathbf{G}_1 p^\alpha, p^\beta, p^\gamma = \begin{bmatrix} p^\alpha + \dfrac{R_{in}}{L} & \dfrac{1}{L} & -\dfrac{1}{L} \\ \\ -\dfrac{1}{C_o} & p^\beta + \dfrac{1}{R_o C_o} & 0 \\ \\ \dfrac{1}{C_b} & 0 & p^\gamma \end{bmatrix} \tag{7a}$$

$$\mathbf{G}_2 = \begin{bmatrix} -\dfrac{R_{in}}{L} & -\dfrac{1}{L} & 0 \\[2mm] \dfrac{1}{C_o} & 0 & 0 \\[2mm] 0 & 0 & \dfrac{1}{R_{in}C_b} \end{bmatrix} \tag{7b}$$

These equations have shown that the fractional orders can only influence the matrix $\mathbf{G}_1$, but not $\mathbf{G}_2$.

In consideration of the ESP method, the equation solution for Eq (6) can be represented by a series

$$\mathbf{x} = \mathbf{x}_0 + \sum_{i=1}^{\infty} \varepsilon^i \mathbf{x}_i \tag{8}$$

where the zero order approximate vector $\mathbf{x}_0$ is called the main wave and $i^{th}$ order approximate vectors $\mathbf{x}_i$ are called the corrections. Smallness indicator parameter $\varepsilon^i$ is introduced temporarily to supply the order of the magnitude in the terms and to indicate that $\mathbf{x}_i$ is much smaller than $\mathbf{x}_0$. More accurately speaking, we have $\mathbf{x}_0 \gg \varepsilon^i \mathbf{x}_i \gg \varepsilon^{i+1} \mathbf{x}_{i+1}$. Similarly, $s(t)$ can be expanded into

$$s(t) = s_0 + \sum_{i=1}^{\infty} \varepsilon^i s_i \tag{9}$$

Substituting Eqs (8) and (9) into $\mathbf{f}(\mathbf{x})$, and merging terms that have the same order $\varepsilon^i$, the expression of $\mathbf{f}(\mathbf{x})$ can be obtained as

$$\mathbf{f} = \mathbf{f}_0 + \sum_{i=1}^{\infty} \varepsilon^i \mathbf{f}_i \tag{10}$$

where

$$\mathbf{f}_0 = s_0 \mathbf{x}_0 \tag{11a}$$

$$\mathbf{f}_1 = s_0 \mathbf{x}_1 + s_1 \mathbf{x}_0 \tag{11b}$$

$$\mathbf{f}_2 = s_0 \mathbf{x}_2 + s_1 \mathbf{x}_1 + s_2 \mathbf{x}_0 \tag{11c}$$

$$\mathbf{f}_3 = s_0 \mathbf{x}_3 + s_1 \mathbf{x}_2 + s_2 \mathbf{x}_1 + s_3 \mathbf{x}_0 \tag{11d}$$

$$\mathbf{f}_4 = s_0 \mathbf{x}_4 + s_1 \mathbf{x}_3 + s_2 \mathbf{x}_2 + s_3 \mathbf{x}_1 + s_4 \mathbf{x}_0 \tag{11e}$$

and so on.

According to Eq (1), the order of the fractional differential operator does not affect the angular frequency. Similar to the method of [15], the terms in expansion (8) can be represented as

$$\mathbf{x}_i = \sum_{m \in E_i} \mathbf{x}_{mi} = \mathbf{a}_{0i} + \sum \left[ \mathbf{a}_{mi} e^{jm\tau} + \bar{\mathbf{a}}_{mi} e^{-jm\tau} \right] \tag{12}$$

Here, $m$ is an integer, the normalized time $\tau = \omega t$ ($\omega = 2\pi/T$). The terms $\mathbf{a}_{0i}$ are the DC components, $\mathbf{a}_{mi}$ represent the $m^{th}$ harmonic magnitudes. The spectral content set of $\{E_0\}$ of the vector $\mathbf{x}_0$, which is a group of numbers denoting relevant frequencies of harmonics, depends on the physical phenomena of object. Considering the low-pass filtering properties of the power converters, the first term $\mathbf{x}_0$ only has the DC components of the state variables, implying that $\mathbf{x}_0 = \mathbf{a}_{00}$ and $\{E_0\} = \{0\}$. Likewise, the spectral content $\{E_i\}$ for each $\mathbf{x}_i$ in the DC-DC converters is unknown in advance. $\{E_i\}$ can be obtained via the iterated operation, which starts with $\mathbf{x}_1$ and moves to higher order corrections.

Like $\mathbf{x}_1$ that uses normalized time $\tau$, the switching function $s(\tau)$ can be depicted in Fourier series as

$$s(\tau) = b_0 + \sum_{l=1}^{\infty}(b_l e^{jl\tau} + \bar{b}_l e^{-jl\tau}) \tag{13}$$

where $b_0 = (\int_0^T s(t)\mathrm{d}t)/T = D$, $b_l = (\alpha_l - j\beta_l)/2$, and $\bar{b}_l$ stands for the conjugate complex of $b_l$, in which

$$\alpha_l = \frac{2}{T}\int_0^T s(t)\cos(l\omega t)dt = \frac{\sin(2Dl\pi)}{l\pi} \tag{14a}$$

and

$$\beta_l = \frac{2}{T}\int_0^T s(t)\sin(l\omega t)dt = \frac{1 - \cos(2Dl\pi)}{l\pi} \tag{14b}$$

The coefficient $b_l$ decreases with the increasing of $l$. According to the definition described in [15], $s_i(\tau)$ can be chosen as

$$s_0(\tau) = b_0 + b_1 e^{j\tau} + \bar{b}_1 e^{-j\tau} \tag{15a}$$

$$s_i(\tau) = b_{2i}e^{j2i\tau} + b_{2i+1}e^{j(2i+1)\tau} + \bar{b}_{2i}e^{-j2i\tau} + \bar{b}_{2i+1}e^{-j(2i+1)\tau} \tag{15b}$$

If Eqs (8) and (9) are introduced to (10), the spectral content of each term $\mathbf{f}_i$ gets wider in comparison with that of $\mathbf{x}_i$. Then, $\mathbf{f}_i$ can be denoted as

$$\mathbf{f}_i = \mathbf{f}_{ik} + \mathbf{R}_{i+1} \tag{16}$$

where

$$\mathbf{f}_{ik} = \sum_{m \in E_i}\mathbf{p}_{mi} = \mathbf{g}_{0i} + \sum(\mathbf{g}_{mi}e^{jm\tau} + \bar{\mathbf{g}}_{mi}e^{-jm\tau}) \tag{17a}$$

$$\mathbf{R}_i = \sum_{m \in E_i}\mathbf{q}_{mi} = \mathbf{V}_{0i} + \sum(\mathbf{V}_{mi}e^{jm\tau} + \bar{\mathbf{V}}_{mi}e^{-jm\tau}) \tag{17b}$$

In the above equations, the spectral content of the term $\mathbf{f}_{0k}$ has the same harmonics with the spectral content of the term $\mathbf{x}_0$, while the additional harmonics outside the set $\{E_0\}$ belong to $\mathbf{R}_1$. Generally speaking, the harmonics of $\mathbf{R}_1$ have magnitudes that are smaller than the magnitudes of the harmonics in $\mathbf{f}_{0k}$. As in [15], the set $\{E_1\}$ is determined by the newly generated harmonics in $\mathbf{f}_0$. In another word, the spectral content of the term $\mathbf{f}_{1k}$ includes the same harmonics with the spectral content of the correction $\mathbf{x}_1$. In the same manner, the spectral contents of the correction $\mathbf{x}_{i+1}$ is determined by the spectral contents in $\mathbf{R}_{i+1}$, and the term $\mathbf{R}_{i+1}$ is

considered smaller compared to $\mathbf{f}_{ik}$ [37]. Thus, Eq (16) can be rewritten as

$$\mathbf{f}_i = \mathbf{f}_{ik} + \varepsilon \mathbf{R}_{i+1} \tag{18}$$

Introducing Eqs (18) into (10), we can obtain that

$$\mathbf{f} = (\mathbf{f}_{0k} + \varepsilon \mathbf{f}_{1k} + \varepsilon^2 \mathbf{f}_{2k} + \cdots) + (\varepsilon \mathbf{R}_1 + \varepsilon^2 \mathbf{R}_2 + \cdots) \tag{19}$$

Then substituting Eqs (8) and (19) into (6) and equating the terms with the same $\varepsilon^i$ on the both sides, we can obtain the following differential equations:

$$\mathbf{G}_1(p^\alpha, p^\beta, p^\gamma)\mathbf{x}_0 + \mathbf{G}_2\mathbf{f}_{0k} = \mathbf{H}_1 + s_0\mathbf{H}_2 \tag{20a}$$

$$\mathbf{G}_1(p^\alpha, p^\beta, p^\gamma)\mathbf{x}_1 + \mathbf{G}_2(\mathbf{f}_{1k} + \mathbf{R}_1) = s_1\mathbf{H}_2 \tag{20b}$$

$$\mathbf{G}_1(p^\alpha, p^\beta, p^\gamma)\mathbf{x}_2 + \mathbf{G}_2(\mathbf{f}_{2k} + \mathbf{R}_2) = s_2\mathbf{H}_2 \tag{20c}$$

$$\mathbf{G}_1(p^\alpha, p^\beta, p^\gamma)\mathbf{x}_3 + \mathbf{G}_2(\mathbf{f}_{3k} + \mathbf{R}_3) = s_3\mathbf{H}_2 \tag{20d}$$

$$\mathbf{G}_1(p^\alpha, p^\beta, p^\gamma)\mathbf{x}_4 + \mathbf{G}_2(\mathbf{f}_{4k} + \mathbf{R}_4) = s_4\mathbf{H}_2 \tag{20e}$$

and so on.

Because the influence of the exponential functions has been eliminated, these equations are all linear. Using the harmonic balance method, the solutions of Eq (20) in the steady state can be found. The amplitudes of the main wave $\mathbf{x}_0$ can be solved by using the Eq (20a). Likewise, the harmonic amplitudes of the corrections $\mathbf{x}_1$, $\mathbf{x}_2$, $\mathbf{x}_3$, $\mathbf{x}_4$,..., etc can be obtained by using Eq (20b) and the following equations.

When the sufficient components are found, the steady-state solution of $\mathbf{x}$ is acquired by a simple summing of these components, which is approximated by

$$\mathbf{x} \approx \mathbf{x}_0 + \mathbf{x}_1 + \mathbf{x}_2 + \mathbf{x}_3 + \mathbf{x}_4 + \cdots \tag{21}$$

Therefore, the parameter $\varepsilon$ was used essentially to point out the order of the equations. The right side of Eqs (Eq (20a))–(20e) can be adjusted according to the order of the exponent function in the left side.

## Solutions for the state variables of the fractional-order positive Luo converter

### Steady-state solutions

In order to conveniently interpret the low-pass filtering characteristics of the power converters, $\mathbf{x}_0$ is selected as

$$\mathbf{x}_0 = \mathbf{a}_{00} = \begin{bmatrix} I_{00} & V_{o00} & V_{b00} \end{bmatrix}^T \tag{22}$$

where $I_{00}$, $V_{o00}$ and $V_{b00}$ are the DC components.

The solutions of $\mathbf{x}_0$, $\mathbf{x}_1$, $\mathbf{x}_2$ in Eqs (Eq (20a))–(20c) can be found in S1 Appendix. Normally, considering the low-pass filtering characteristic of the power converters, the magnitudes of harmonics with order higher than three are quite small. This represents that, for most of DC-DC converters, the solution of $\mathbf{x}$ in the steady state can be approximated by $\mathbf{x}_0$, $\mathbf{x}_1$ and $\mathbf{x}_2$. However, for the positive Luo converter, $v_b$ varies greatly when the switch $S$ is on. In other words, the magnitudes of its high frequency harmonic components are very high

correspondingly. The high order correction $\mathbf{x}_i (i \geq 3)$ can greatly affect the low order harmonic magnitudes of $\mathbf{x}$, especially for the magnitudes of the DC component and the exponent $e^{-j\tau}$. Increasing the order of the correction $\mathbf{x}_i$ we consider can greatly improve the accuracy of $\mathbf{x}$. Take $\mathbf{x}_3$ and $\mathbf{x}_4$ as examples, their DC components and the first order harmonic magnitudes are relatively larger. Therefore, these terms cannot be ignored.

According to S1 Appendix, the spectral content set of $\mathbf{x}_3$ can be derived as $\{E_3\} = \{1, 4, 5\}$. The correction $\mathbf{x}_3$ can be assumed to be

$$\mathbf{x}_3 = \mathbf{a}_{13} e^{j\tau} + \bar{\mathbf{a}}_{13} e^{-j\tau} + \mathbf{a}_{43} e^{j4\tau} + \bar{\mathbf{a}}_{43} e^{-j4\tau} \mathbf{a}_{53} e^{j5\tau} + \bar{\mathbf{a}}_{53} e^{-j5\tau} \quad (23)$$

in which $\mathbf{a}_{13} = [I_{13}\ V_{o13}\ V_{b13}]^T$ gives the corrections of the first order harmonics in $\mathbf{a}_{11}$. By introducing $s_i$ and $\mathbf{x}_i (i = 0, 1, 2, 3)$ into $\mathbf{f}_3$, the following expressions of $\mathbf{f}_{3m}$ and $\mathbf{R}_4$ can be concluded.

$$
\begin{aligned}
\mathbf{f}_{3m} = \ & (b_0\mathbf{a}_{13} + b_3\bar{\mathbf{a}}_{22} + \bar{b}_2\mathbf{a}_{32})e^{j\tau} + (b_0\mathbf{a}_{43} + \bar{b}_1\mathbf{a}_{53} + b_2\mathbf{a}_{22} + b_5\bar{\mathbf{a}}_{11})e^{j4\tau} \\
& + (b_0\mathbf{a}_{53} + b_1\mathbf{a}_{43} + b_3\mathbf{a}_{22} + b_2\mathbf{a}_{32} + b_4\mathbf{a}_{11})e^{j5\tau} + c.c
\end{aligned} \quad (24a)
$$

$$
\begin{aligned}
\mathbf{R}_4 = \ & (b_1\bar{\mathbf{a}}_{13} + \bar{b}_1\mathbf{a}_{13} + \bar{b}_2\mathbf{a}_{22} + b_2\bar{\mathbf{a}}_{22} + \bar{b}_3\mathbf{a}_{32} + b_3\bar{\mathbf{a}}_{32}) + (b_1\mathbf{a}_{13} + b_2\mathbf{a}_{02})e^{j2\tau} \\
& + (\bar{b}_1\mathbf{a}_{43} + b_3\mathbf{a}_{02} + b_4\bar{\mathbf{a}}_{11})e^{j3\tau} + (b_1\mathbf{a}_{53} + b_3\mathbf{a}_{32} + b_5\mathbf{a}_{11} + b_6\mathbf{a}_{00})e^{j6\tau} + c.c
\end{aligned} \quad (24b)
$$

Substituting $\mathbf{x}_3$, $\mathbf{f}_{3m}$ and $\mathbf{R}_3$, the following equation can be obtained:

$$(\mathbf{G}_{11} + \mathbf{G}_2 b_0)\mathbf{a}_{13} = -\mathbf{G}_2(b_3\bar{\mathbf{a}}_{22} + \bar{b}_2\mathbf{a}_{32} + b_1\mathbf{a}_{02} + \bar{b}_1\mathbf{a}_{22} + b_2\bar{\mathbf{a}}_{11}) \quad (25)$$

Eq (25) can be overwritten as

$$
\left( \begin{bmatrix} (j\omega)^\alpha + \dfrac{R_{in}}{L} & \dfrac{1}{L} & -\dfrac{1}{L} \\[2ex] -\dfrac{1}{C_o} & (j\omega)^\beta + \dfrac{1}{R_o C_o} & 0 \\[2ex] \dfrac{1}{C_b} & 0 & (j\omega)^\gamma \end{bmatrix} + D \begin{bmatrix} -\dfrac{R_{in}}{L} & -\dfrac{1}{L} & 0 \\[2ex] \dfrac{1}{C_o} & 0 & 0 \\[2ex] 0 & 0 & \dfrac{1}{R_{in} C_b} \end{bmatrix} \right) \begin{bmatrix} I_{13} \\[2ex] V_{o13} \\[2ex] V_{b13} \end{bmatrix}
$$

$$
= - \begin{bmatrix} -\dfrac{R_{in}}{L} & -\dfrac{1}{L} & 0 \\[2ex] \dfrac{1}{C_o} & 0 & 0 \\[2ex] 0 & 0 & \dfrac{1}{R_{in} C_b} \end{bmatrix} \begin{bmatrix} b_3\bar{I}_{22} + \bar{b}_2 I_{32} + b_1 I_{02} + \bar{b}_1 I_{22} + b_2 \bar{I}_{11} \\[2ex] b_3\bar{V}_{o22} + \bar{b}_2 V_{o32} + b_1 V_{o02} + \bar{b}_1 V_{o22} + b_2 \bar{V}_{o11} \\[2ex] b_3\bar{V}_{b22} + \bar{b}_2 V_{b32} + b_1 V_{b02} + \bar{b}_1 V_{b22} + b_2 \bar{V}_{b11} \end{bmatrix}
$$

$$(26)$$

Thus, $\mathbf{a}_{13}$ is obtained. From Eq (24b), the spectral content set of $\mathbf{x}_4$ can be deduced, $\{E_4\} = \{0, 2, 3, 6\}$. Then the correction $\mathbf{x}_4$ can be assumed to be

$$\mathbf{x}_4 = \mathbf{a}_{04} + \mathbf{a}_{24} e^{j2\tau} + \bar{\mathbf{a}}_{24} e^{-j2\tau} + \mathbf{a}_{34} e^{j3\tau} + \bar{\mathbf{a}}_{34} e^{-j3\tau} + \mathbf{a}_{64} e^{j6\tau} + \bar{\mathbf{a}}_{64} e^{-j6\tau} \quad (27)$$

where $\mathbf{a}_{04} = [I_{04}\ V_{o04}\ V_{b04}]^T$ gives the corrections of the DC components in $\mathbf{a}_{00}$, similarly.

Introducing $s_i$ and $\mathbf{x}_i (i = 0, 1, 2, 3, 4)$ into $\mathbf{f}_4$, the following expression of $\mathbf{f}_{4m}$ can be deduced.

$$\mathbf{f}_{4m} = \quad b_0\mathbf{a}_{04} + (b_0\mathbf{a}_{24} + \bar{b}_1\mathbf{a}_{34} + \bar{b}_2\mathbf{a}_{43} + b_3\bar{\mathbf{a}}_{13} + \bar{b}_3\mathbf{a}_{53} + b_4\bar{\mathbf{a}}_{22})e^{j2\tau}$$
$$+ (b_0\mathbf{a}_{34} + b_1\mathbf{a}_{24} + b_2\mathbf{a}_{13} + \bar{b}_2\mathbf{a}_{53})e^{j3\tau} + (b_0\mathbf{a}_{64} + b_2\mathbf{a}_{43} + b_4\mathbf{a}_{22} + b_7\bar{\mathbf{a}}_{11})e^{j6\tau} + c.c \tag{28}$$

Utilizing $\mathbf{x}_4$, $\mathbf{f}_{4m}$ and $\mathbf{R}_4$, we can obtain the following equation:

$$(\mathbf{G}_{10} + \mathbf{G}_2 b_0)\mathbf{a}_{04} = -\mathbf{G}_2(b_1\bar{\mathbf{a}}_{13} + \bar{b}_1\mathbf{a}_{13} + \bar{b}_2\mathbf{a}_{22} + b_2\bar{\mathbf{a}}_{22} + \bar{b}_3\mathbf{a}_{32} + b_3\bar{\mathbf{a}}_{32}) \tag{29}$$

Eq (29) can be represented in the matrix form as

$$\left( \begin{bmatrix} \dfrac{R_{in}}{L} & \dfrac{1}{L} & -\dfrac{1}{L} \\[2ex] -\dfrac{1}{C_o} & \dfrac{1}{R_o C_o} & 0 \\[2ex] \dfrac{1}{C_b} & 0 & 0 \end{bmatrix} + D \begin{bmatrix} -\dfrac{R_{in}}{L} & -\dfrac{1}{L} & 0 \\[2ex] \dfrac{1}{C_o} & 0 & 0 \\[2ex] 0 & 0 & \dfrac{1}{R_{in}C_b} \end{bmatrix} \right) \begin{bmatrix} I_4 \\[2ex] V_{o4} \\[2ex] V_{b4} \end{bmatrix}$$

$$= - \begin{bmatrix} -\dfrac{R_{in}}{L} & -\dfrac{1}{L} & 0 \\[2ex] \dfrac{1}{C_o} & 0 & 0 \\[2ex] 0 & 0 & \dfrac{1}{R_{in}C_b} \end{bmatrix} \begin{bmatrix} b_1\bar{I}_{13} + \bar{b}_1 I_{13} + \bar{b}_2 I_{22} + b_2\bar{I}_{22} + b_3\bar{I}_{32} + \bar{b}_3 I_{32} \\[2ex] b_1\bar{V}_{o13} + \bar{b}_1 V_{o13} + \bar{b}_2 V_{o22} + b_2\bar{V}_{o22} + \bar{b}_3 V_{o32} + b_3\bar{V}_{o32} \\[2ex] b_1\bar{V}_{b13} + \bar{b}_1 V_{b13} + \bar{b}_2 V_{b22} + b_2\bar{V}_{b22} + \bar{b}_3 V_{b32} + b_3\bar{V}_{b32} \end{bmatrix} \tag{30}$$

In this manner, $\mathbf{a}_{04}$ is solved. Furthermore, the approximate steady state solution of $\mathbf{x}$ can be expressed as

$$\mathbf{x} \quad \approx \mathbf{x}_0 + \mathbf{x}_1 + \mathbf{x}_2 + \mathbf{x}_3 + \mathbf{x}_4 + \cdots$$
$$= (\mathbf{a}_{00} + \mathbf{a}_{02} + \mathbf{a}_{04}) + (\mathbf{a}_{11} + \mathbf{a}_{13})e^{j\tau} + \mathbf{a}_{22}e^{j2\tau} + \mathbf{a}_{32}e^{j3\tau} + c.c + \cdots \tag{31}$$

Components of $\mathbf{x}$ are

$$i_L \approx \quad (I_{00} + I_{02} + I_{04}) + 2[real(I_{11} + I_{13})\cos\omega t - imag(I_{11} + I_{13})\sin\omega t$$
$$+ real(I_{22})\cos2\omega t - imag(I_{22})\sin2\omega t \tag{32a}$$
$$+ real(I_{32})\cos3\omega t - imag(I_{32})\sin3\omega t] + \cdots$$

$$v_o \approx \quad (V_{o00} + V_{o02} + V_{o04}) + 2[real(V_{o11} + V_{o13})\cos\omega t - imag(V_{o11} + V_{o13})\sin\omega t$$
$$+ real(V_{o22})\cos2\omega t - imag(V_{o22})\sin2\omega t \tag{32b}$$
$$+ real(V_{o32})\cos3\omega t - imag(V_{o32})\sin3\omega t] + \cdots$$

$$v_b \approx \quad (V_{b00} + V_{b02} + V_{b04}) + 2[real(V_{b11} + V_{b13})\cos\omega t - imag(V_{b11} + V_{b13})\sin\omega t$$
$$+ real(V_{b22})\cos2\omega t - imag(V_{b22})\sin2\omega t \tag{32c}$$
$$+ real(V_{b32})\cos3\omega t - imag(V_{b32})\sin3\omega t] + \cdots$$

where $real(\bullet)$ and $imag(\bullet)$ stand for the real part and the imaginary part of complex terms, respectively.

## Transient solutions

As high order oscillation equations have little effect on the transient process, we should focus on the main expression of the state variables, which can be presented as:

$$p^{\alpha} i_{L0} = -\frac{(1-d)R_{in}}{L} i_{L0} - \frac{(1-d)}{L} v_{o0} + \frac{1}{L} v_{b0} + \frac{(1-d)}{L} V_{in} \tag{33a}$$

$$p^{\beta} v_{o0} = \frac{(1-d)}{C_o} i_{L0} - \frac{1}{R_o C_o} v_{o0} \tag{33b}$$

$$p^{\gamma} v_{b0} = -\frac{1}{C_b} i_{L0} - \frac{d}{R_{in} C_b} v_{b0} + \frac{1}{R_{in} C_b} V_{in} \tag{33c}$$

Using the explicit Grünwald-Letnikov (GL) approximation, the numerical solution of Eq (33) has the following form:

$$
\begin{aligned}
i_{L0}(t_k) &= h^{\alpha} \left[ -\frac{(1-d)R_{in}}{L} i_{L0}(t_{k-1}) - \frac{(1-d)}{L} v_{o0}(t_{k-1}) + \frac{1}{L} v_{b0}(t_{k-1}) + \frac{(1-d)}{L} V_{in} \right] \\
&\quad + \sum_{i=1}^{k} w_i^{(\alpha)} i_{L0}(t_{k-i}) + r_k^{(\alpha)} i_{L0}(0)
\end{aligned}
\tag{34a}
$$

$$
\begin{aligned}
v_{o0}(t_k) &= h^{\beta} \left[ \frac{(1-d)}{C_o} i_{L0}(t_k) - \frac{1}{R_o C_o} v_{o0}(t_{k-1}) \right] + \sum_{i=1}^{k} w_i^{(\beta)} v_{o0}(t_{k-i}) \\
&\quad + r_k^{(\beta)} v_{o0}(0)
\end{aligned}
\tag{34b}
$$

$$
\begin{aligned}
v_{b0}(t_k) &= h^{\gamma} \left[ -\frac{1}{C_b} i_{L0}(t_k) - \frac{d}{R_{in} C_b} v_{b0}(t_{k-1}) + \frac{1}{R_{in} C_b} V_{in} \right] + \sum_{i=1}^{k} w_i^{(\gamma)} v_{b0}(t_{k-i}) \\
&\quad + r_k^{(\gamma)} v_{b0}(0)
\end{aligned}
\tag{34c}
$$

where $t_k = kh$, $h$ represents the step time size. $i_{L0}(t_k)$ and $v_{o0}(t_k)$ are the main instantaneous components of the inductor current and the output voltage at time $t_k$ respectively. The coefficients can be calculated by the following expressions:

$$w_l^{(\lambda)} = -\frac{\Gamma(l-\lambda)}{\Gamma(-\lambda)\Gamma(l+1)} \tag{35a}$$

$$r_k^{(\lambda)} = \frac{1}{k^{\lambda} \Gamma(1-\lambda)} \tag{35b}$$

where $\Gamma(\bullet)$ represents the gamma function. $r_k^{(\lambda)}$ is necessary to improve the accuracy in the case of a homogeneous initial value.

To obtain the high order harmonics of the transient solutions, results of Eqs (34a)–(34c) can replace the main wave $\mathbf{a}_{00}$. And then they will be substituted back to the formulas (Eq (20a))–(20e) in Section 4. The approximate transient solutions of the state variables can be expressed as the sum of all the harmonics, the same as Eq (32).

## Comparison and simulation of the conventional schemes and the proposed method

### Comparison of the steady-state solutions

In order to get the analytical solutions of the fractional-order DC-DC converters in the steady state, there are many methods proposed by previous researchers. Among them, the predictor-corrector method [38] is the most widely used used numerical scheme based on the fractional-order definition and the time-domain analysis. It can solve the fractional-order equations in each operating state. By running the routines programmed in *MATLAB*, the numerical analysis of the fractional-order DC-DC converters can be accomplished cycle by cycle. Through this process, the final results would include the complete contents of the state variables from the initial state to the stable state. On the other hand, the whole information can also be obtained by performing *MATLAB/Simulink* simulations with the modified Oustaloup's approximation method [39]. The fractional integral module in the simulation is replaced by the approximated equations. As mentioned in the introduction, the modified Oustaloup's approximation method is a precise engineering simulation running in the frequency domain. Therefore, a modified Oustaloup's approximation method proposed in this section will be used to verify the validity of the proposed method, and be used as the comparison criterion for other methods.

In this subsection, the circuit parameters of the fractional-order positive Luo converter are listed as follows: the input voltage $V_{in}$ = 10 *V*, the switch frequency $f$ = 20 *kHz*, the duty ratio $D$ = 0.5, $L$ = 1 *mH*, $C_b$ = 47 *μF*, $C_o$ = 10 *μF*, and $R_o$ = 50 Ω. This part compares the results of the modified Oustaloup's approximation method, the PECE-ABM method and the proposed scheme. The accuracy of the state variables is the main comparing object. In the modified Oustaloup's method, there are three key parameters: the filter order $2N + 1$, the lower limit $\omega_b$ and the upper limit $\omega_h$ of the fitting frequency. Normally, $\omega_b \omega_h$ = 1. We choose $\omega_h = 5 \times 10^5$ *rad/s*, $\omega_b = 2 \times 10^{-6}$ *rad/s* and $N$ = 8 for the fractional-order positive Luo converter in this paper. The results gotten by these three methods are listed in Table 1.

From Table 1, we can see that the DC components of these three schemes keep good consistency. Evidently, both $i_L$ and $v_o$ are dependent of the orders of the capacitor and the inductor. No matter which method is selected, the DC components of the state variables decrease with the decreasing of $\alpha$, $\beta$ and $\gamma$. In order to further compare these three methods, the steady state ripples are shown in Figs 3 and 4, where the black dash lines, the blue dotted lines and the red solid lines stand for the results of the PECE-ABM method, the modified Oustaloup's approximation method and the proposed scheme, respectively.

As shown in Figs 3 and 4, the waveforms from these three schemes are consistent with each other, and the steady state ripples are influenced by the fractional orders. Specifically, the steady state ripples of $i_L$ and $v_o$ increase with the decreasing of $\alpha$ and $\beta$ respectively. In general, the steady state ripples gotten by the proposed scheme are more closely resemble to the simulation results obtained by the modified Oustaloup's method, especially with smaller values of $\alpha$ and $\beta$. For the PECE-ABM method, the rangeabilities of both the DC components and the AC components are all undersized when $\alpha$ and $\beta$ changes. This proves that the proposed method can track the dramatic changes more closely compared to the PECE-ABM method. Reducing the step size can improve the accuracy of the PECE-ABM, but the number of iterative computations will be greatly increased at the same time. For the modified Oustaloup's method, although it is more accurate, complex fractional-order component approximation and circuit simulation should be used to obtain the steady state solutions.

In order to show that the proposed method can obtain more accurate state variables, of the positive Luo converters, compared to the simplified equivalent small parameter (SESP)

**Table 1. The DC parts of $I_L$ and $V_o$ acquired by using different schemes.**

| $(\alpha, \beta, \gamma)$ | Proposed Method ($a_{00}a_{02}a_{04}$) | | PECE Method | | Oustaloup's Method | |
|---|---|---|---|---|---|---|
| | $I_L$ (A) | $V_o$ (V) | $I_L$ (A) | $V_o$ (V) | $I_L$ (A) | $V_o$ (V) |
| (1, 1, 1) | 1.1696 | 29.2919 | 1.1585 | 29.2847 | 1.1784 | 29.5141 |
| (1, 1, 0.95) | 1.1618 | 29.1003 | 1.1585 | 29.0356 | 1.1677 | 29.2579 |
| (1, 0.9, 1) | 1.1545 | 28.9929 | 1.1622 | 29.0623 | 1.1647 | 29.2449 |
| (1, 0.9, 0.95) | 1.1467 | 28.8024 | 1.1519 | 28.8151 | 1.1541 | 28.9926 |
| (1, 0.8, 1) | 1.0873 | 27.4978 | 1.1107 | 27.9653 | 1.1012 | 27.8534 |
| (1, 0.8, 0.95) | 1.0799 | 27.3154 | 1.1012 | 27.7348 | 1.0916 | 27.6223 |
| (0.95, 1, 1) | 1.1760 | 29.2553 | 1.1703 | 29.2268 | 1.1829 | 29.4770 |
| (0.95, 1, 0.95) | 1.1681 | 29.0621 | 1.1591 | 28.9665 | 1.1716 | 29.2138 |
| (0.95, 0.9, 1) | 1.1568 | 28.9126 | 1.1631 | 28.9278 | 1.1650 | 29.1632 |
| (0.95, 0.9, 0.95) | 1.1489 | 28.7200 | 1.1520 | 28.6693 | 1.1539 | 28.9040 |
| (0.95, 0.8, 1) | 1.0799 | 27.2967 | 1.1043 | 27.6246 | 1.0911 | 27.6438 |
| (0.95, 0.8, 0.95) | 1.0724 | 27.1108 | 1.0941 | 27.3836 | 1.0809 | 27.4064 |
| (0.9, 1, 1) | 1.1951 | 29.1875 | 1.1781 | 29.1446 | 1.1973 | 29.4103 |
| (0.9, 1, 0.95) | 1.1872 | 28.9923 | 1.1656 | 28.8689 | 1.1849 | 29.1343 |
| (0.9, 0.9, 1) | 1.1692 | 28.7659 | 1.1708 | 28.7388 | 1.1724 | 29.0157 |
| (0.9, 0.9, 0.95) | 1.1613 | 28.5703 | 1.1585 | 28.4638 | 1.1602 | 28.7440 |
| (0.9, 0.8, 1) | 1.0774 | 26.9268 | 1.1031 | 27.1367 | 1.0815 | 27.2600 |
| (0.9, 0.8, 0.95) | 1.0698 | 26.7354 | 1.0919 | 26.8795 | 1.0705 | 27.0108 |

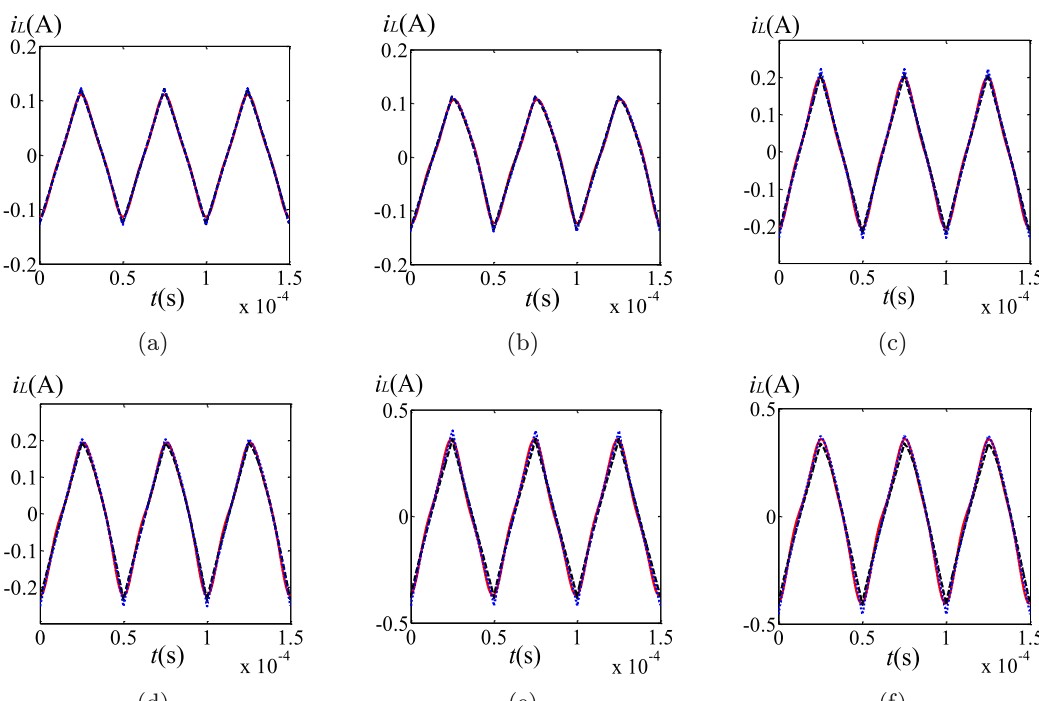

(a)        (b)        (c)

(d)        (e)        (f)

**Fig 3. Comparison of the steady state ripples in $i_L$ under different fractional orders:** (a) $(\alpha, \beta, \gamma) = (1, 1, 0.95)$; (b) $(\alpha, \beta, \gamma) = (1, 0.8, 0.95)$; (c) $(\alpha, \beta, \gamma) = (0.95, 1, 0.95)$; (d) $(\alpha, \beta, \gamma) = (0.95, 0.8, 0.95)$; (e) $(\alpha, \beta, \gamma) = (0.9, 1, 0.95)$; (f) $(\alpha, \beta, \gamma) = (0.9, 0.8, 0.95)$.

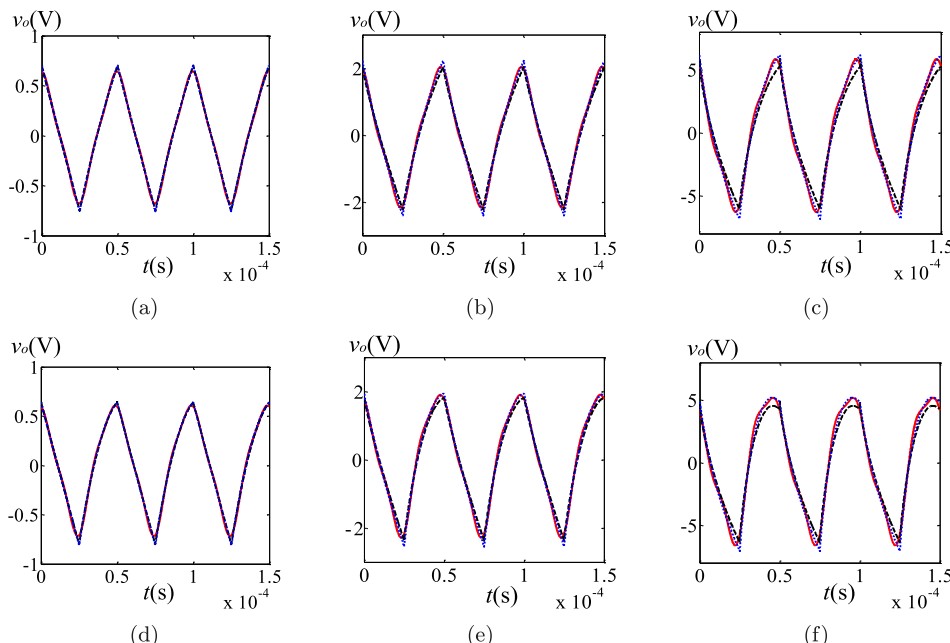

**Fig 4. Comparison of the steady state ripples in $v_o$ under different fractional orders: (a) $(\alpha, \beta, \gamma) = (1, 1, 0.95)$; (b) $(\alpha, \beta, \gamma) = (1, 0.9, 0.95)$; (c) $(\alpha, \beta, \gamma) = (1, 0.8, 0.95)$; (d) $(\alpha, \beta, \gamma) = (0.9, 1, 0.95)$; (e) $(\alpha, \beta, \gamma) = (0.9, 0.9, 0.95)$; (f) $(\alpha, \beta, \gamma) = (0.9, 0.8, 0.95)$.**

method in [15, 16], the tolerance error index [40] is used to indicate the end of the iteration process. This index is the ratio of the matrix-norm of $\mathbf{a}_{mi}$ ($m^{th}$ harmonic magnitude of $\mathbf{X}_i$) to that of $\mathbf{a}_{00}$. Thus, the tolerance of $\mathbf{a}_{mi}$ is denoted as

$$Tami = \frac{\|\mathbf{a}_{mi}\|}{\|\mathbf{a}_{00}\|} \times 100\% \tag{36}$$

The 1% tolerance error index is utilized to control the iteration process of $\mathbf{x}_i$. The correcting process ends once the tolerance error index is smaller than 1%. For most DC-DC converters, $\mathbf{x}$ in the steady state approximating by $\mathbf{x}_0$, $\mathbf{x}_1$ and $\mathbf{x}_2$ is accurate enough. However, for the positive Luo converter, with the decreasing of $\beta$, the tolerance error index of $\mathbf{a}_{32}$ can reach 1% or even larger, as represented in Fig 5, implying that the harmonic magnitudes in the high order correction $\mathbf{x}_i(i \geq 3)$ are correspondingly high compared to the DC component. Thus, the iteration process of $\mathbf{x}_i$ should be continued. Fig 5 also shows the tolerance error index of $\mathbf{a}_{13}$ and $\mathbf{a}_{04}$. This index of $\mathbf{a}_{13}$ is inversely proportional to $\alpha$ and $\beta$, and can be close to 1%. Because the computational efforts for calculating the amplitude of the first three order harmonic amplitude are relatively small, this paper only computes $\mathbf{a}_{13}$ and $\mathbf{a}_{04}$. As depicted in Fig 5, the tolerance error index of $\mathbf{a}_{04}$ with different fractional orders keeps below 0.6%. Therefore, the approximate solution of $\mathbf{x}$ calculated by the proposed method is precise enough for the positive Luo converter shown in Fig 1. And the tolerance error index at the end of the iteration process decreases from larger than 1.3% to 0.6%.

In order to comprehensively compare the steady state solutions obtained by the SESP method, the Oustaloup's approximation method and the proposed scheme, we consider the case in which $(\alpha, \beta, \gamma) = (0.9, 0.8, 0.95)$. Fig 6 shows the comparison of these three schemes, which obviously shows that the steady state solutions obtained by the proposed method can be more closely resemble to the results gotten by the Oustaloup's approximation method,

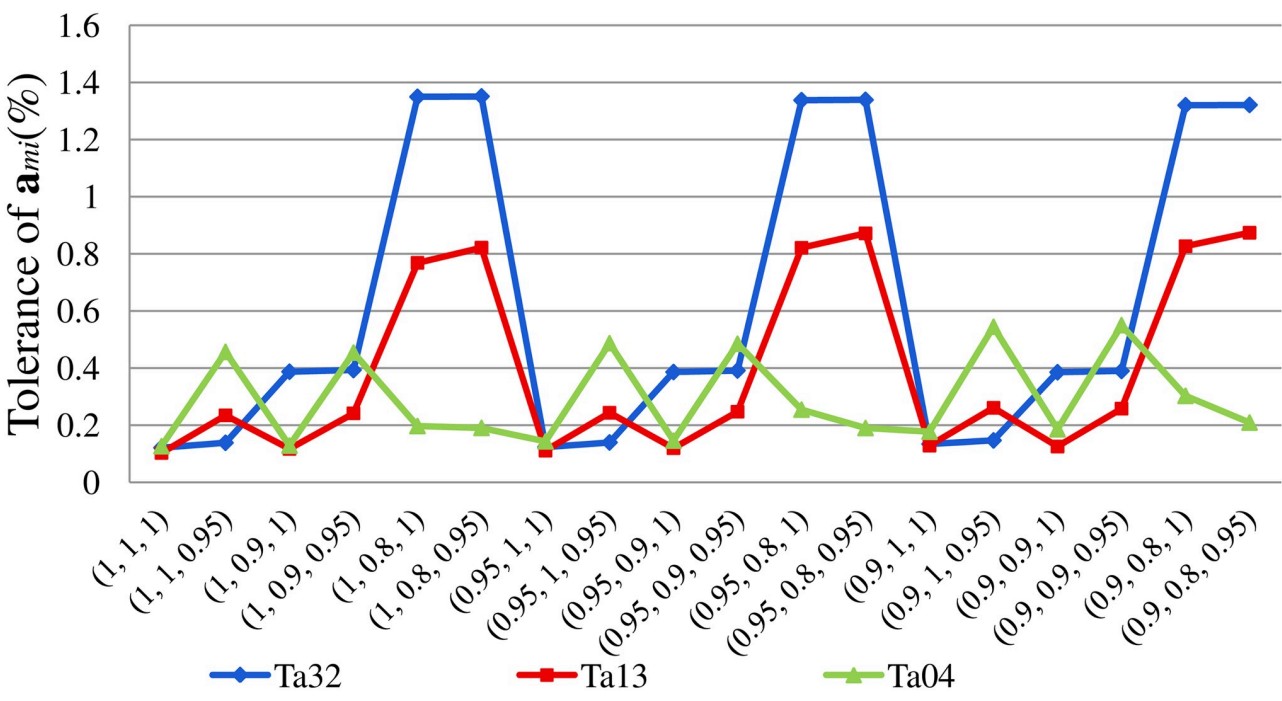

**Fig 5. The tolerance error index of $m^{th}$ harmonic magnitude in $x_i$ ($a_{32}$, $a_{13}$ and $a_{04}$).**

compared to the SESP method, especially for $v_b$. When the power switch turns off, the gap between the results gotten by the SESP method and the proposed scheme can be greater than 3%. Similar to the tolerance error index of $\mathbf{a}_{mi}$, the tolerance error indexes of each state variable are calculated. It can be obtain that $\mathbf{Ta13} = [Ti_{L13}\ Tv_{o13}\ Tv_{b13}]^T = [1.81\%\ 0.88\%\ 0.76\%]^T$ and $\mathbf{Ta04} = [Ti_{L04}\ Tv_{o04}\ Tv_{b04}]^T = [0.02\%\ 0.01\%\ 0.67\%]^T$. For $v_b$, the lack of significant decrease in tolerance error index indicates that the number of iterations for SESP method is not enough. Thus, the corrections of DC value and main wave proposed in this paper are necessary.

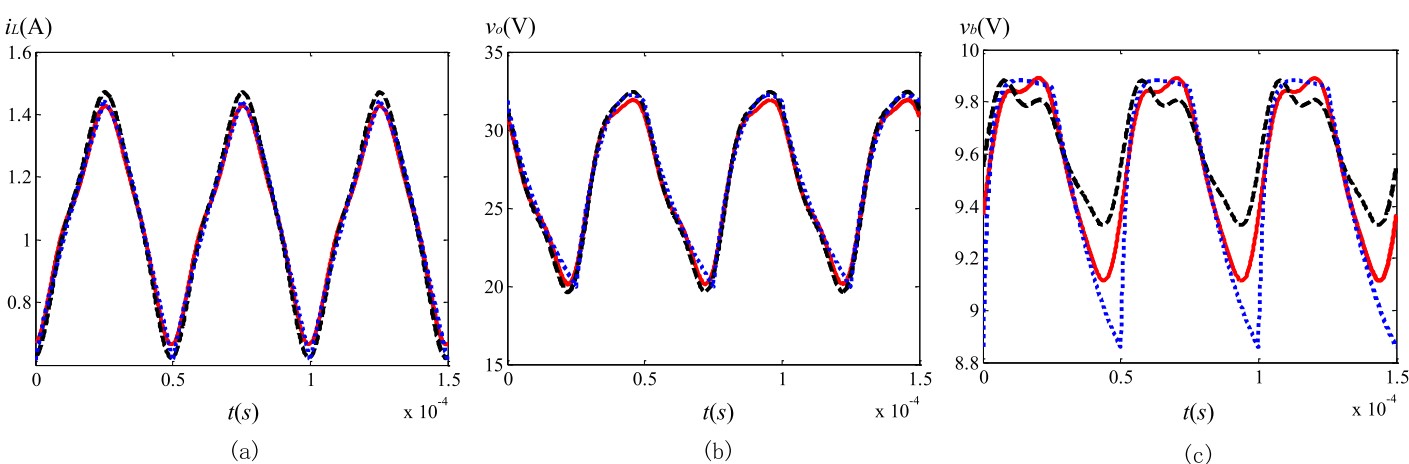

**Fig 6. The steady state solutions obtained by using the SESP method (black dash lines), the proposed method (red solid lines) and the Oustaloup's approximation method (blue solid lines): (a) $i_L$; (b) $v_o$; (c) $v_b$.**

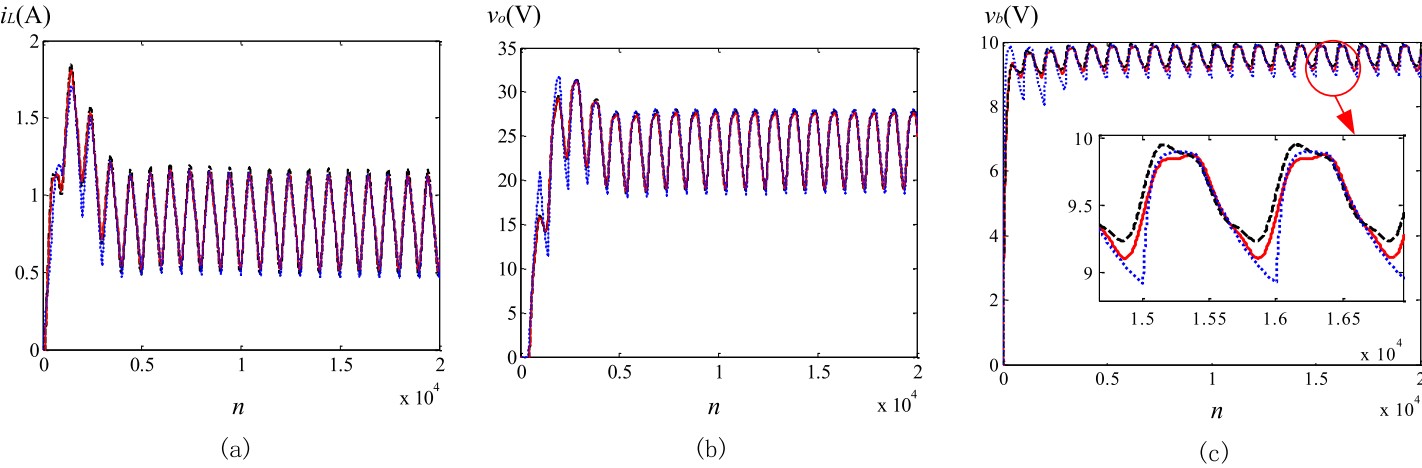

**Fig 7. Transient waveforms comparison of solutions obtained by the SESP method (black dash lines), the proposed method (red solid lines) and the Oustaloup's approximation method (blue solid lines): (a) $i_L$; (b) $v_o$; (c) $v_b$.**

### Transient solutions comparison

To verify the transient solutions obtained with the proposed method, a comparison is made between them and other simulation results obtained by using the SESP method and the modified Oustaloup's method. Circuit parameters in this subsection are listed as follows: $V_{in}$ = 10 V, $f$ = 20 kHz, $D$ = 0.4, $L$ = 1 mH, $C_b$ = 47 μF, $C_o$ = 10 μF, $R_o$ = 50 Ω, $(\alpha, \beta, \gamma)$ = (0.9, 0.8, 0.95).

The comparison of the state variables versus $n$ ($n = (1000f)^{-1} t$), between the SESP method, the Oustaloup's approximation method and the proposed scheme, is depicted in Fig 7. It can be seen that, the transient solutions of the proposed method are in good accordance with those from the Oustaloup's approximation method. Based on the analysis of the previous subsection, using the proposed method, the numerical solution of each cycle is more accurate than SESP method. Because the explicit Grünwald-Letnikov (GL) approximation is used in the transient solution calculation, this error will accumulate with the increase of iterations, and be more apparent in the steady state. Therefore, the steady state solutions obtained by the proposed method are more accurate compared to the SESP method, especially for $v_b$. When the power switch turns on, the gap between the SESP method and the proposed method is more obvious than that shown in Fig 6. The error between the maximum values of $v_b$ obtained by the two methods can exceed 1%. Moreover, the $v_b$ waveform obtained by the SESP method during the switching on conduction appears obvious distortion, which does not conform to the actual situation. Expect for the increased accuracy, the number of iterations is greatly reduced using proposed method, since the step size is not required to be smaller than the switching cycle.

## Analysis of fractional order related phenomena
### CCM operating boundary

According to the operating criterion of the positive Luo converter, when the circuit of Fig 1 operates in continuous-conduction mode (CCM), the current $i_L$ must be continuous, meaning that the inductor current is always greater or equal to zero. The CCM operating criterion ccan be obtained from two parameters, namely $\bar{I}_L$ and $\Delta i_L$, which denote the average value of the inductor current and the peak-peak ripple, respectively. $\bar{I}_L$ can be approximated by the combination of $I_{00}$, $I_{02}$ and $I_{04}$. $\Delta i_L$ can be computed using the inductor volt-second balance principle

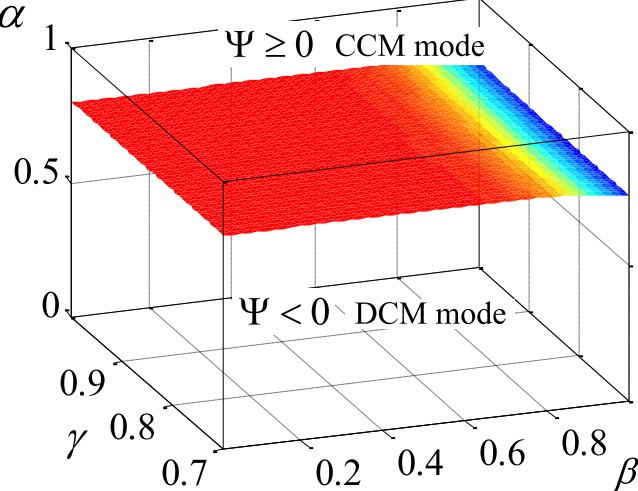

**Fig 8. Operating criterion of fractional-order positive Luo converter.**

and the definition of fractional-order derivative given by Caputo. As $R_{in}$ is a very small value, $v_b$ is very close to the input voltage $V_{in}$ during the ON-state of the switch. Due to this and Eq (4a), the CCM operating formula of the positive Luo converter with the fractional order can be concluded as

$$\Psi = \bar{I}_L - \frac{1}{2}\Delta i_L \approx I_{00} + I_{02} + I_{04} - \frac{V_{in}(DT)^{\alpha}}{2L\alpha\Gamma(\alpha)} \qquad (37)$$

where $\Gamma(\bullet)$ represents the gamma function. The fractional-order positive Luo converter is operating in CCM mode when $\Psi \geq 0$. In other words, the converter goes to discontinuous conduction mode (DCM) when $\Psi < 0$.

According to the calculation method derived in the previous Section, the 3D plot of the CCM operating criterion can be found in Fig 8 utilizing the circuit parameters in subsection A of Section 4. The space beneath the surface in Fig 8 indicate the DCM mode, while the above part represents the CCM mode. As shown in this figure, the CCM operating criterion is mainly determined by the order $\alpha$, while $\beta$ partly affects it, and the impact of $\gamma$ on the boundary can be neglected.

## Transfer functions of the converter

The transfer functions of the converter can be obtained by the small-signal perturbation analysis based on Eq (6). Perturbed values can be represented as

$$\tilde{\mathbf{x}} = \mathbf{X} + \hat{\mathbf{x}} \qquad (38a)$$

$$\tilde{d} = D + \hat{d} \qquad (38b)$$

$$\tilde{v}_{in} = V_{in} + \hat{v}_{in} \qquad (38c)$$

where $\tilde{\mathbf{x}}$, $\tilde{d}$ and $\tilde{v}_{in}$ represent three perturbations. And $\mathbf{X}$, $D$ and $V_{in}$ are the DC parts

corresponding to the perturbed values. Substituting Eqs (38a)–(38c) into (6) gives expression

$$[\mathbf{G}_1(p^\alpha, p^\beta, p^\gamma) + \mathbf{G}_2 \tilde{d}]\tilde{\mathbf{x}} = (\mathbf{H}_1 + \tilde{d}\mathbf{H}_2)\tilde{v}_{in} \tag{39}$$

Neglecting the infinitely small $2^{th}$ perturbation can yield the following equation

$$\mathbf{G}_1(p^\alpha, p^\beta, p^\gamma)\hat{\mathbf{x}} + \mathbf{G}_2(D\hat{\mathbf{x}} + \hat{d}\mathbf{X}) = \mathbf{H}_1\hat{v}_{in} + \mathbf{H}_2(D\hat{v}_{in} + \hat{d}V_{in}) \tag{40}$$

In order to get the transfer functions from $\hat{v}_{in}$ to $\hat{\mathbf{x}}$, $\hat{d} = 0$ should be assumed. Hence, Eq (40) can be changed into

$$[\mathbf{G}_1(p^\alpha, p^\beta, p^\gamma) + \mathbf{G}_2 D]\hat{\mathbf{x}} = (\mathbf{H}_1 + \mathbf{H}_2 D)\hat{v}_{in} \tag{41}$$

Then, $\hat{\mathbf{x}}$ is depicted as

$$
\begin{aligned}
\hat{\mathbf{x}} &= [\mathbf{G}_1(p^\alpha, p^\beta, p^\gamma) + \mathbf{G}_2 D]^{-1}(\mathbf{H}_1 + \mathbf{H}_2 D)\hat{v}_{in} \\
&= \left(\begin{bmatrix} p^\alpha + \dfrac{(1-D)R_{in}}{L} & \dfrac{1-D}{L} & -\dfrac{1}{L} \\[2ex] -\dfrac{1-D}{C_o} & p^\beta + \dfrac{1}{R_o C_o} & 0 \\[2ex] \dfrac{1}{C_b} & 0 & p^\gamma + \dfrac{D}{R_{in} C_b} \end{bmatrix}\right)^{-1} \begin{bmatrix} \dfrac{(1-D)V_{in}}{L} \\[2ex] 0 \\[2ex] \dfrac{DV_{in}}{R_{in} C_b} \end{bmatrix} \hat{v}_{in}
\end{aligned}
\tag{42}
$$

Similarly, by letting $\hat{v}_{in} = 0$, the transfer functions from $\hat{d}$ to $\hat{\mathbf{x}}$ can be obtained. Thus, Eq (40) can be rewritten as

$$\mathbf{G}_1(p^\alpha, p^\beta, p^\gamma)\hat{\mathbf{x}} + \mathbf{G}_2(D\hat{\mathbf{x}} + \hat{d}\mathbf{X}) = \mathbf{H}_2\hat{d}V_{in} \tag{43}$$

where $\mathbf{X} \approx \mathbf{a}_{00}$. In this case, $\hat{\mathbf{x}}$ is denoted by

$$
\begin{aligned}
\hat{\mathbf{x}} &= [\mathbf{G}_1(p^\alpha, p^\beta, p^\gamma) + \mathbf{G}_2 D]^{-1}(\mathbf{H}_2\hat{d}V_{in} - \mathbf{G}_2\hat{d}\mathbf{X}) \\
&= \left(\begin{bmatrix} p^\alpha + \dfrac{(1-D)R_{in}}{L} & \dfrac{1-D}{L} & -\dfrac{1}{L} \\[2ex] -\dfrac{1-D}{C_o} & p^\beta + \dfrac{1}{R_o C_o} & 0 \\[2ex] \dfrac{1}{C_b} & 0 & p^\gamma + \dfrac{D}{R_{in} C_b} \end{bmatrix}\right)^{-1} \\[2ex]
&\quad \times \left(\begin{bmatrix} -\dfrac{1}{L} \\[2ex] 0 \\[2ex] \dfrac{1}{R_{in} C_b} \end{bmatrix} V_{in} - \begin{bmatrix} -\dfrac{R_{in}}{L} & -\dfrac{1}{L} & 0 \\[2ex] \dfrac{1}{C_o} & 0 & 0 \\[2ex] 0 & 0 & \dfrac{1}{R_{in} C_b} \end{bmatrix} \begin{bmatrix} I_0 \\[2ex] V_{o0} \\[2ex] V_{b0} \end{bmatrix}\right) \hat{d}
\end{aligned}
\tag{44}
$$

In this way, the final transfer functions are equal to the results gotten with the state-space averaging model. However, the derivation is much simpler. From the Fig 9(a) and 9(c), it can

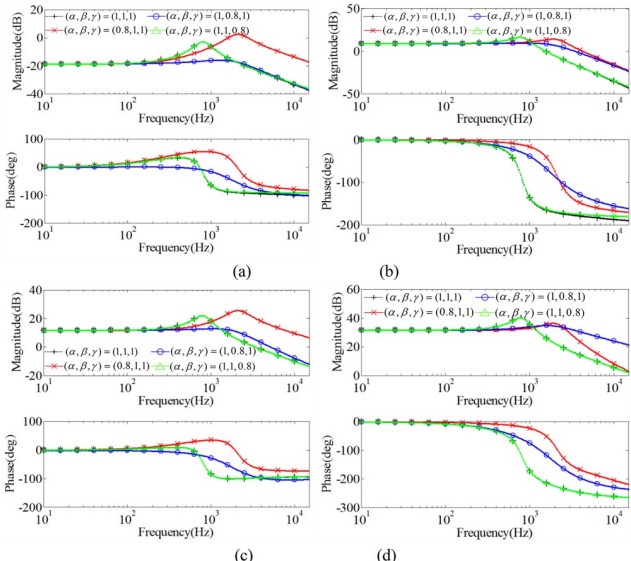

**Fig 9. Bode diagrams of $G_{i_L v_{in}}(s)$, $G_{v_o v_{in}}(s)$, $G_{iL}d(s)$ and $G_{vo}d(s)$ with different $\alpha$, $\beta$ and $\gamma$: (a) $G_{i_L v_{in}}(s)$; (b) $G_{v_o v_{in}}(s)$; (c) $G_{iL}d(s)$; (d) $G_{vo}d(s)$.**

be seen that $\alpha$ significantly affects $G_{i_L v_{in}}(s)$ and $G_{iL}d(s)$ with the same fractional order. With the decrease of $\alpha$, the open loop amplitude margin increases. In Fig 9(b) and 9(d), both $\alpha$ and $\beta$ can affect $G_{v_o v_{in}}(s)$ and $G_{v0}d(s)$. While, the order $\gamma$ has very little influence on the bode diagrams.

### Harmonics of the state variables

Based on the results gotten by the proposed scheme, we further analyze the harmonics of the inductor current $i_L$ and the output voltage $v_o$ with different orders. In a word, with the decreasing of $\alpha$, $\beta$ and $\gamma$, the RMS values of the harmonics increase as displayed in Fig 10. This directly results in the increasing of the inductor current ripples and the output voltage ripples, which are generally undesired in the DC-DC converters design. Separately, $\alpha$ has a significant impact on the first and third harmonics of $i_L$, while $\beta$ exerts considerable influence on the first and third harmonics of $v_o$. However, their effects on the second harmonic are reversed. The order $\alpha$ decreases with the increasing of the second harmonic amplitude in $v_o$. Similarly, $\beta$ decreases

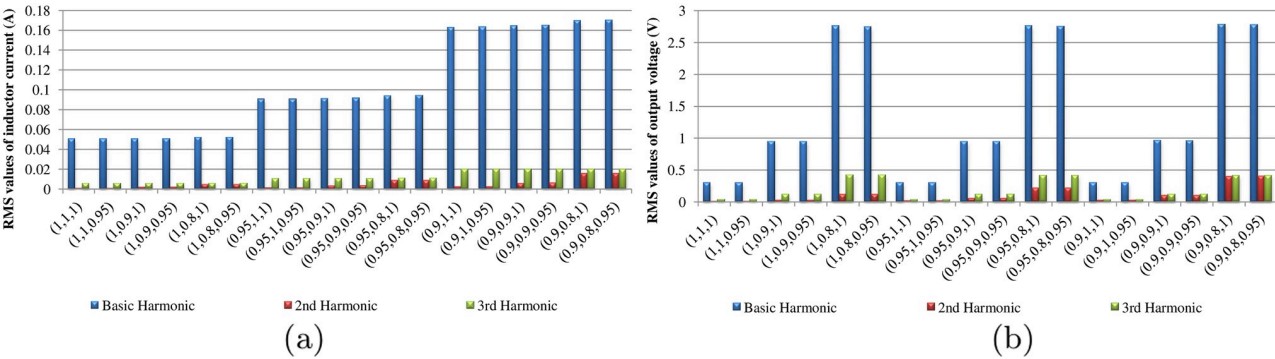

**Fig 10. RMS values of harmonics in $i_L$ and $v_o$ with different fractional orders: (a) $i_L$; (b) $v_o$.**

when the second harmonic amplitude of $i_L$ increases. Normally, the order $\gamma$ has very little influence on $i_L$ and $v_o$, relatively speaking. It mainly affects $i_L$ by influencing the value of $v_b$.

## Dynamical behavior analysis of the PI controlled fractional-order positive Luo converter

In this section, whether the non-existence isolated period oscillations can be observed in fractional-order model is used to establish the validity of proposed model in analyzing dynamical behavior.

In order to analyze dynamical behaviors in the fractional-order model, the block diagram of the positive Luo converter with PI voltage compensator are described in Fig 11. The control voltage $v_{vf}$ can be expressed as

$$v_{vf} = k_p(v_{ref} - k_v v_o) + k_I \int (v_{ref} - k_v v_o)dt \tag{45}$$

The integral coefficient $k_I$ of the controller is treated as the bifurcation parameter, which is critical in the practical design. The component values and the control parameters of the test bench, as shown in Fig 11, are chosen as: $V_{in} = 10\ V$, $f = 20\ kHz$, $L = 4\ mH$, $C_b = 47\ \mu F$, $C_o = 10\ \mu F$, $R_o = 50\ \Omega$, $V_{ref} = 18\ V$, $k_v = 1$, $k_p = 0.01$.

With the parameter $k_I$ varying from 10 to 110, the bifurcation graphs of the integer-order and the fractional-order circuits are depicted in Fig 12, which are only constructed with the steady state data at the beginning of each switching cycle. Take the integer-order system as an example. As presented in Fig 12(a), the converter remains stable in period-1 orbit when $k_I$ varies from 10 to 34. Once $k_I$ reaches to 34, the Hopf bifurcation appears, forming the bifurcation points. Fig 12(a)–12(d) have shown that the bifurcation points are shifting backward with the decreasing fractional order $\alpha$ of converters, given a fixed $\beta$ value. In other words, with the same parameters, the bifurcation occurs in the integer-order circuit, while the fractional-order converter remains asymptotically stable. Similarly, the value of $k_I$ at the time of bifurcation increases when $\beta$ decreases, which effect is dominating. The analysis discussed above shows that the fractional-order converter is easier to keep stable.

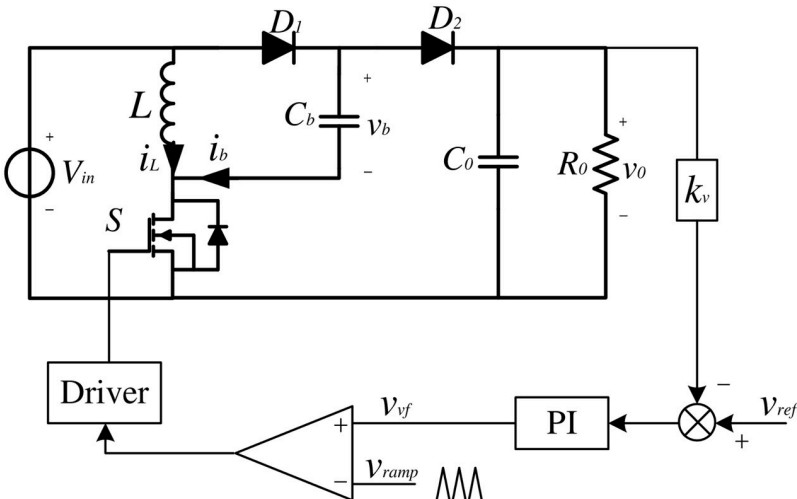

**Fig 11. Block diagram showing the positive Luo converter with PI voltage compensator.**

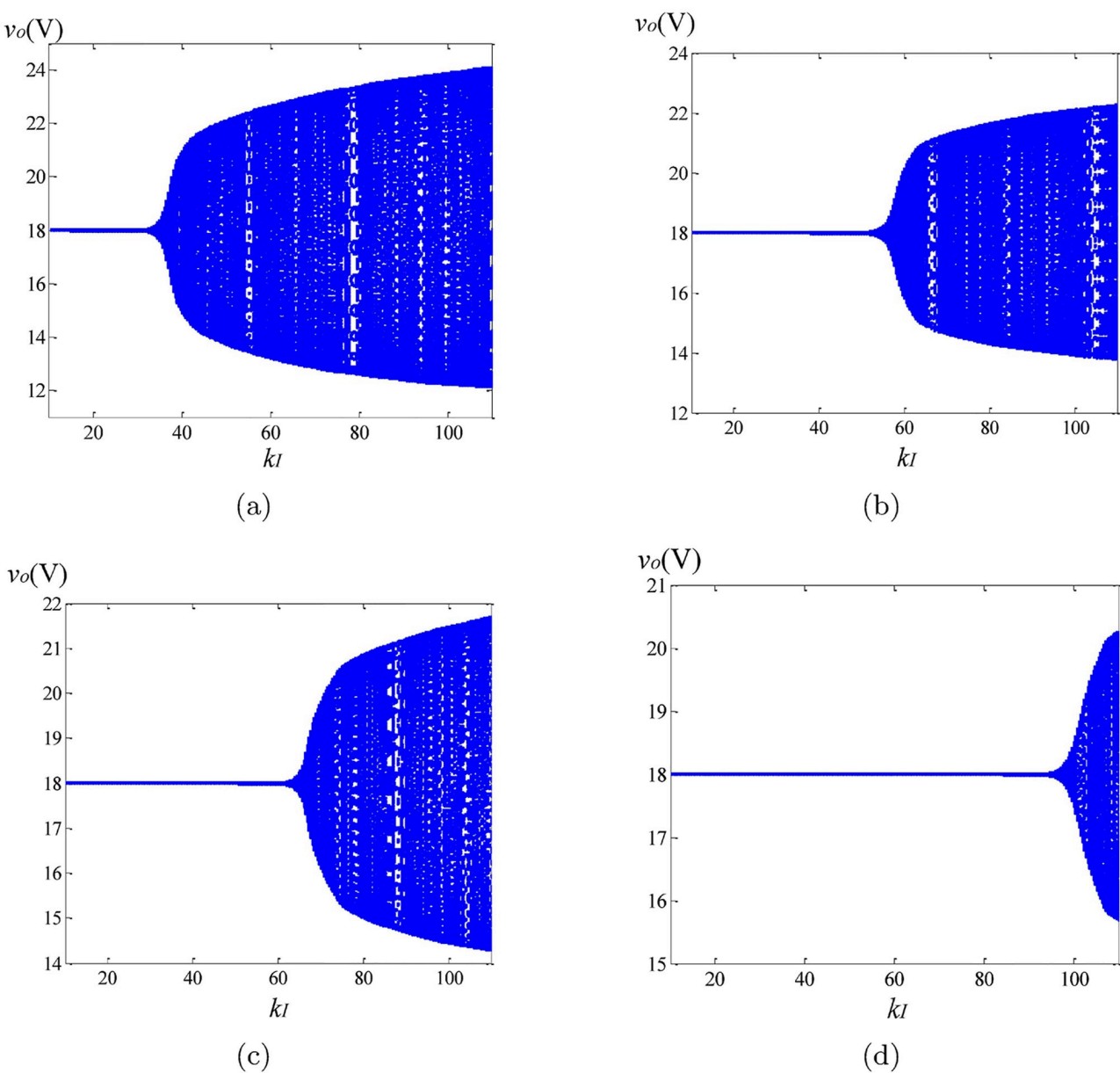

**Fig 12. Bifurcation diagrams ($v_o$ vs. $k_I$) of the integer-order and the fractional-order converters:** (a) $(\alpha, \beta, \gamma) = (1, 1, 1)$; (b) $(\alpha, \beta, \gamma) = (0.95, 1, 1)$; (c) $(\alpha, \beta, \gamma) = (1, 0.95, 1)$; (d) $(\alpha, \beta, \gamma) = (0.95, 0.95, 1)$.

To observe the causes of the asymptotically limit cycle behavior, we keep $k_I = 105$. To reveal the differences of the non-linear behaviors between the fractional and the integer order converters, the phase portraits, the harmonic spectrums and the time-domain curves are utilized. The simulation results can be found in Figs 13–15. Because the steady state solutions in the proposed model are acquired without using the definitions of fractional calculus, the vital difference between the essence of regular oscillations in the integer and the fractional order systems can be evinced. Unlike their integer order corresponding systems under the same condition, the limit cycle behaviors do not exist in the fractional-order converters. Take the fractional-order converter with $(\alpha, \beta, \gamma) = (0.95, 0.95, 1)$ as an example. Fig 15(b) illustrates the

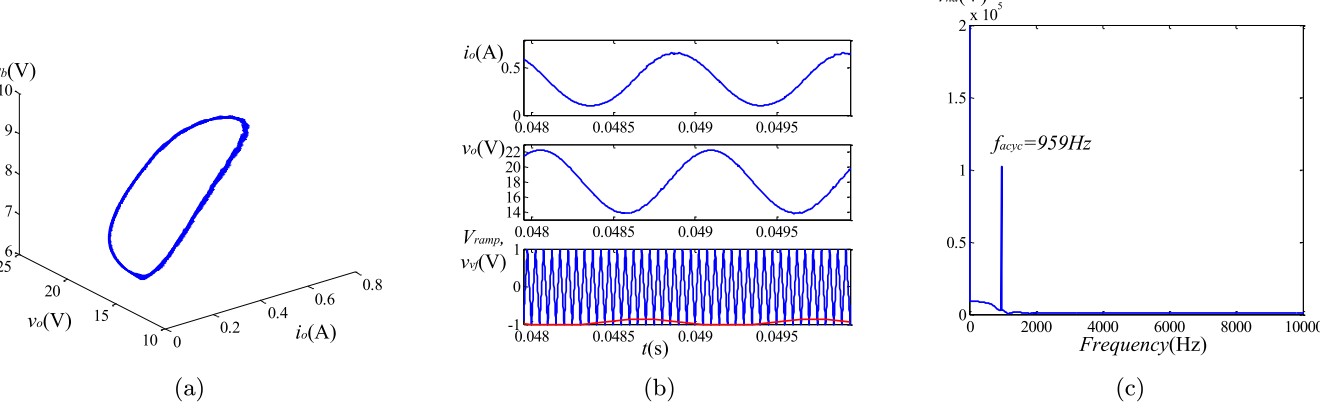

**Fig 13. Simulation results of asymptotically limit cycle behavior with (α, β, γ) = (0.95, 1, 1): (a) phase portrait; (b) time-domain waveform; (c) harmonic spectrum.**

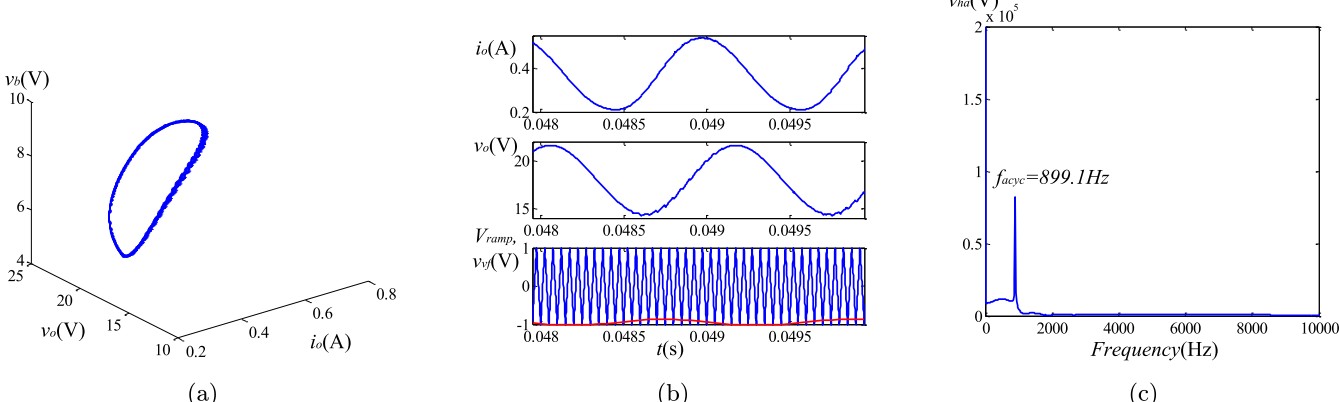

**Fig 14. Simulation results of asymptotically limit cycle behavior with (α, β, γ) = (1, 0.95, 1): (a) phase portrait; (b) time-domain waveform; (c) harmonic spectrum.**

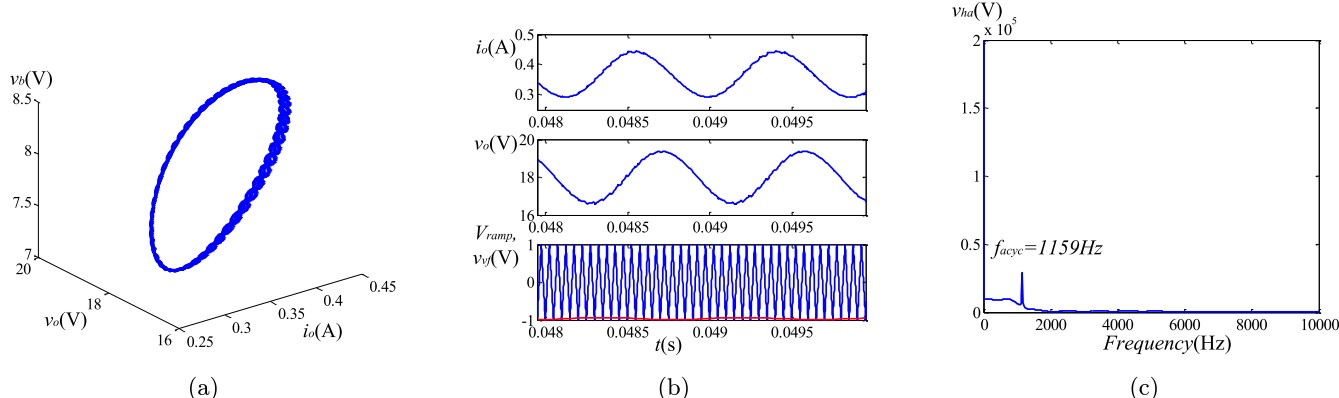

**Fig 15. Simulation results of asymptotically limit cycle behavior with (α, β, γ) = (0.95, 0.95, 1): (a) phase portrait; (b) time-domain waveform; (c) harmonic spectrum.**

time-domain waveforms of the output voltage $v_o$, the inductor current $i_L$, the control voltage $v_{vf}$ and the triangular wave $v_{ramp}$. It can be seen that in the steady state, the control signal $v_{vf}$ (red line) reaches the lower limit of $v_{ramp}$ (blue line) several times, but the inductor current $i_L$ remains continuous all the time. The phase graph (Fig 15(a)) seems to be a trajectory of the periodic oscillation (solar to the limit cycle), while actually this oscillation is not perfectly periodic but is generating non-periodic items obviously. If the approximation of the fractional-order components in [13] is used, the phase graph will be a closed curve like Fig 13(a), which does not satisfy the well-known fact that the fractional-order systems cannot have any periodic solutions. Fig 15(c) shows the most significant harmonic component occurs at a frequency of 1159 Hz, indicating the frequency of the asymptotically limit cycle $f_{acyc}$ under this condition. It is lower than the switching frequency. From above analysis and the plots in Figs 13–15, it can be revealed that $f_{acyc}$ increases with the decreasing of the fractional order. Besides, the influence of $\beta$ is more obvious. In addition, since the transient solutions are obtained using the explicit Grünwald-Letnikov (GL) approximation, the limit cycle behaviors, which normally appear in the time-domain analysis, do not exist in the converters when using the proposed method. The only existence of the asymptotically period oscillation can further demonstrate the validity of the proposed modeling of the non-linear behaviors.

## Circuit simulation and experimental results

As there are no commercially available fractional-order components, many schemes are designed to build the equivalent circuit of the fractional-order components. In order to further analyze the simulation results listed in previous section, according to the method in [41], the approximation circuits of the inductor and the capacitor with fractional-order properties are built by using the resistor/inductor or resistor/capacitor networks (as shown in Fig 16(a) and 16(b) respectively). The parameters in the approximation circuit of the fractional-order components are deduced in S2 Appendix. And the circuit parameters are the same with the simulation in subsection A of Section 4.

### Circuit simulation results

Let $L$, $C_b$ and $C_o$ in Fig 1 be replaced by the fractional-order inductor and capacitor units shown in Fig 16(a) and 16(b) correspondingly, the circuit simulation results can be found in Figs 17 and 18. All these simulations are accomplished by *PLECS Standalone*.

For example, the bode diagrams of $G_{v_o v_{in}}(s)$ and $G_{vo}d(s)$ with $(\alpha, \beta, \gamma)$ = (0.9, 0.9, 0.95) and $(\alpha, \beta, \gamma)$ = (0.9, 0.8, 0.95) respectively are shown in Fig 17. These curves are obtained by the theoretical analysis in Section 3 and the *PLECS* circuit simulation separately. The theoretical

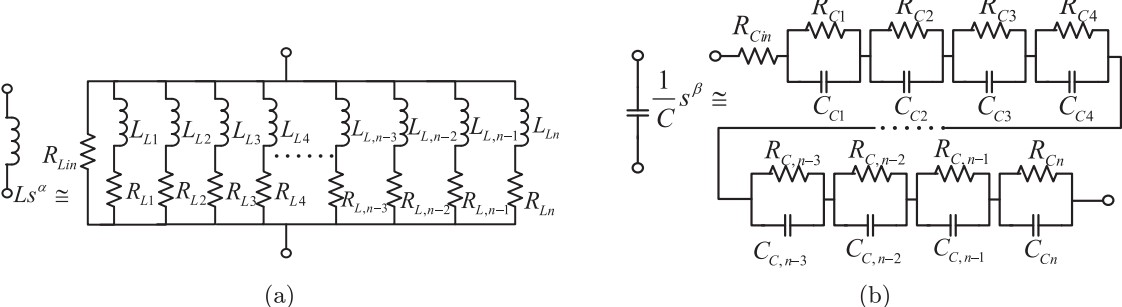

(a)                                                                              (b)

**Fig 16. The approximate circuit of components with fractional-order property: (a) fractional-order inductor; (b) fractional-order capacitor.**

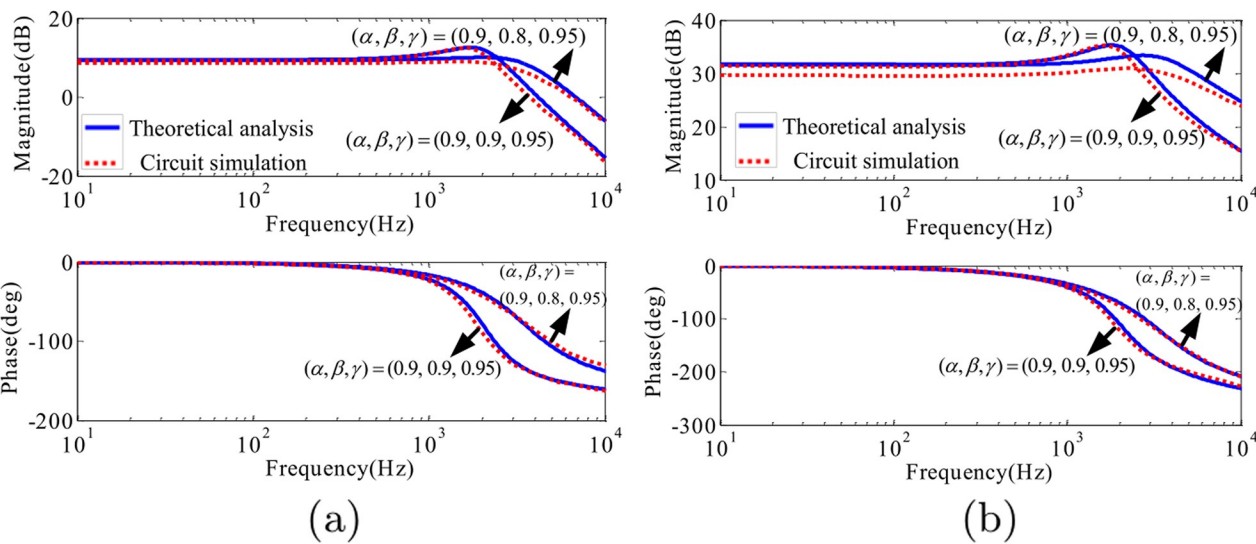

**Fig 17. Bode diagram obtained by the theoretical analysis and the circuit simulation with ($\alpha$, $\beta$, $\gamma$) = (0.9, 0.9, 0.95) and ($\alpha$, $\beta$, $\gamma$) = (0.9, 0.8, 0.95): (a) $G_{v_o v_{in}}(s)$; (b) $G_{vo}d(s)$.**

equations of $G_{v_o v_{in}}(s)$ and $G_{vo}d(s)$ gotten by the proposed fractional-order model have been expressed by (42) and (44). As depicted in Fig 17, the circuit simulation results are basically like the theoretical analysis. The discrepancies are mainly caused by the approximation of circuit for the fractional-order inductor and capacitor.

As shown in the steady state waveforms in Fig 18, with decreasing of the fractional-order $\alpha$ and $\beta$, the ripple of $i_L$ and $v_o$ increase significantly, which is consistent with the results of the theoretical analysis and the numerical simulation in the previous sections.

Especially, as shown in Fig 18(c), when $\alpha$ = 0.8, the converter enters into the DCM mode, in which $v_o$ changes greatly. All of these phenomena agree very well with the analysis of the CCM operating criterion, as shown in Fig 8.

To verify the mechanism of the asymptotically limit cycle behavior, a circuit simulation of the positive Luo converter with the PI voltage compensator (schematically presented in Fig 11) has been performed. The simulation setup has the same parameters as listed in Section 6. Fig 19 shows the time-domain waveforms of the inductor current and the output voltage when $k_I$ = 105. The green and the red curves present the simulation results of the converters with ($\alpha$, $\beta$, $\gamma$) = (1, 1, 1) and ($\alpha$, $\beta$, $\gamma$) = (0.95, 0.95, 1) respectively. As shown in Fig 19, the waveforms have a good agreement with the numerical simulation results described in Section 6.

## Experimental results

In order to further verify the effectiveness of the proposed method and to realize the practical performance of fractional-order components, a positive Luo converter prototype is built. Fig 20 shows the photograph of the positive Luo converter. In this prototype, power switch and diode in Fig 1 choose STB18N20 and MUR160 respectively. To reduce the interference of inductors in the equivalent implementation circuit of fractional-order inductor, some shielded techniques, such as shielded power inductors are utilized.

Fig 21 shows the steady-state wave forms of $i_L$ and $v_o$, which is obtained by experimental test with different fractional orders. Experimental measurements of these state variables are listed in Table 2. From Fig 21(a)–21(d) and Table 2, it can be seen that the RMS and peak-peak

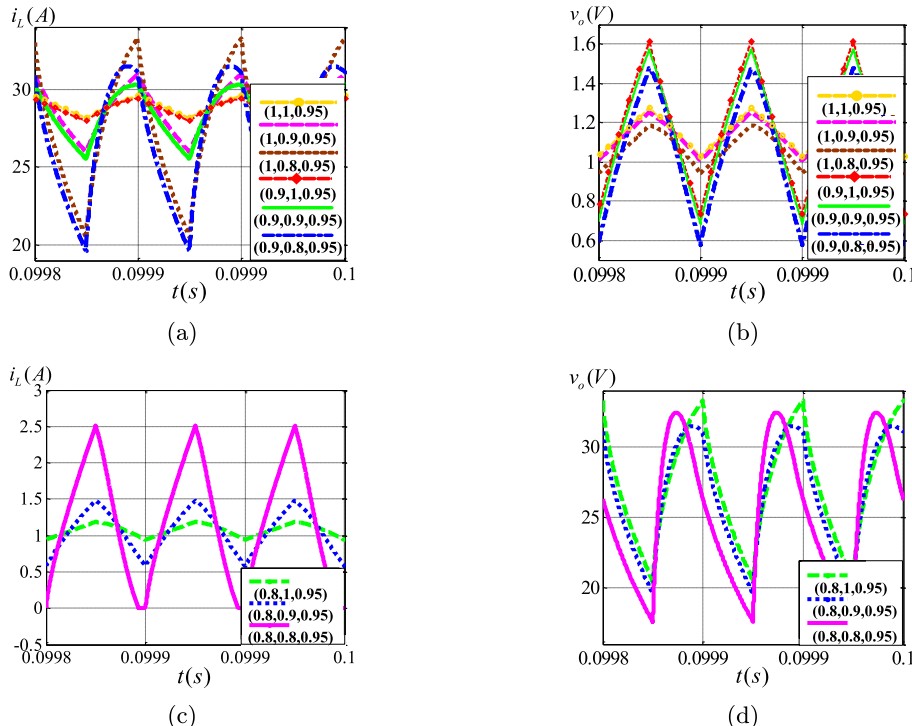

**Fig 18. State variables in the steady state under different fractional orders: (a) $i_L$ in CCM; (b) $v_o$ in CCM; (c) $i_L$ in CCM and DCM; (d) $v_o$ in CCM and DCM.**

value of $i_L$ and $v_o$ are basically consistent with the simulation analysis. Considering the influence of parasitic parameters and electromagnetic interference, the error between them can be ignored.

## Conclusion

A modified fractional-order modeling and asymptotically limit cycle behavior analysis method for the positive Luo converters is proposed in this paper. With the proposed method, the approximate steady state solutions can be obtained without utilizing the definitions of

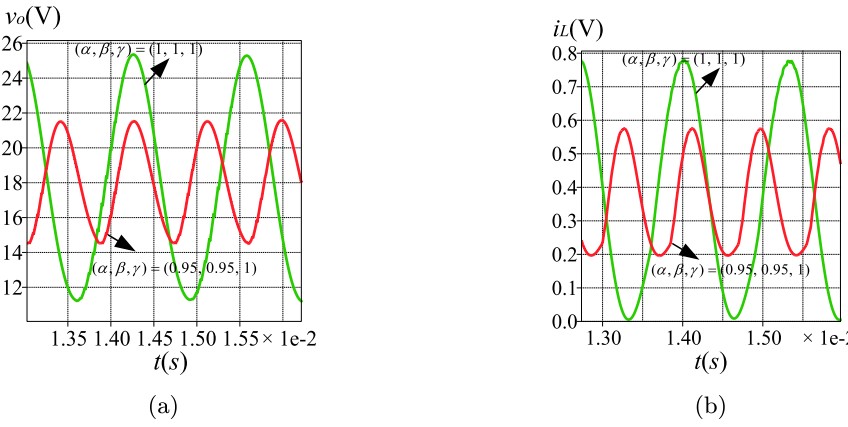

**Fig 19. Circuit simulation results of (a) $v_o$; (b) $i_L$.**

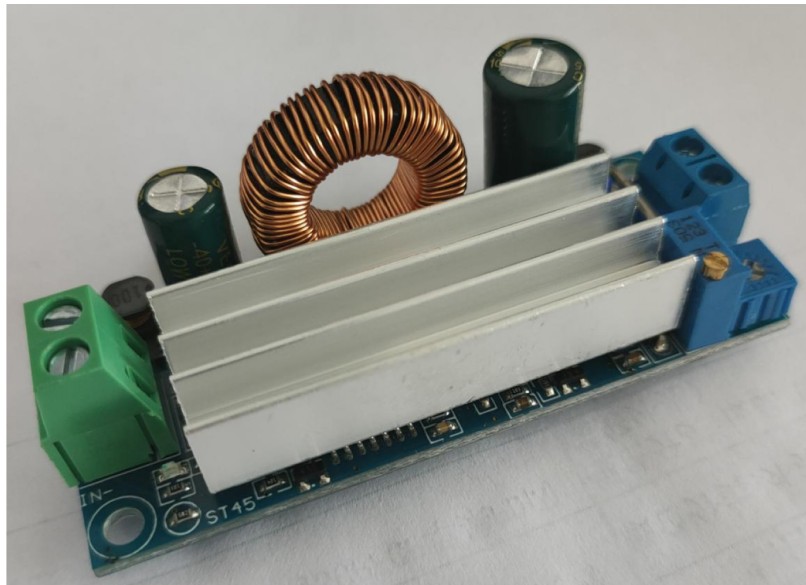

**Fig 20. Photograph of the prototype positive Luo converter.**

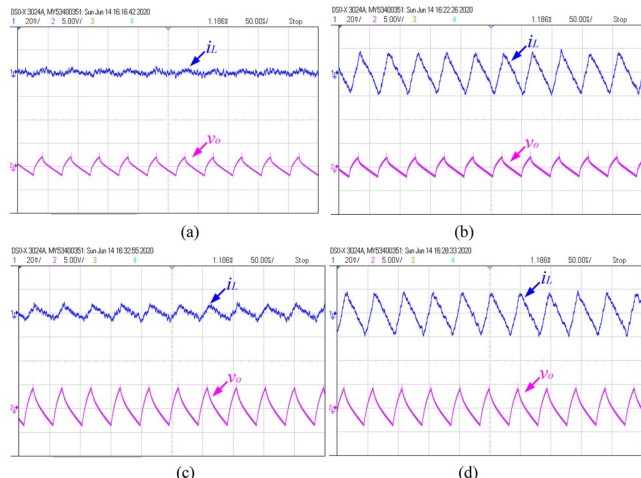

**Fig 21. Experimental results under different fractional orders: (a) $(\alpha, \beta, \gamma) = (1, 1, 1)$; (b) $(\alpha, \beta, \gamma) = (0.9, 1, 1)$; (c) $(\alpha, \beta, \gamma) = (1, 0.9, 1)$; (d) $(\alpha, \beta, \gamma) = (0.9, 0.9, 1)$.**

**Table 2. Experimental results under different fractional orders.**

| $(\alpha, \beta, \gamma)$ | $i_{L\_RMS}$ | $\Delta i_{L\_pp}$ | $v_{o\_RMS}$ | $\Delta v_{o\_pp}$ |
|---|---|---|---|---|
| (1, 1, 1) | 1.112 | 0.062 | 29.42 | 1.83 |
| (0.9, 1, 1) | 1.131 | 0.098 | 29.23 | 1.87 |
| (1, 0.9, 1) | 1.096 | 0.196 | 29.16 | 4.79 |
| (0.9, 0.9, 1) | 1.097 | 0.198 | 28.95 | 4.84 |

fractional calculus. In addition, it eliminates the needs for the circuit simulations or multiple iterations. Based on the particularity of the positive Luo converter, the improved algorithm can ensure the accuracy without adding more harmonics in the approximate expression, compared to [32]. The final transient solution can be acquired by replacing the DC part by the primary transient components. Therefore, the proposed modeling methodology provides a more convenient and faster approach to the optimal design of the fractional-order positive Luo converters. Furthermore, the proposed method, with the time-domain model using the explicit Grünwald-Letnikov (GL) approximation, can analyze the nonlinear behaviors via the numerical analysis more accurately. The analysis of the dynamic behaviors in this paper confirms that nonexistence of the periodic solutions in continuous-time fractional-order is a remarkable difference between the integer-order converters and the fractional-order converters.

As the DC parts and the harmonics of the state variables in the converters are directly related to the fractional orders, changing orders influences the characteristics in the steady state. The most obvious phenomenon is that the ripples of the state variables are markedly depended on the fractional orders. More specifically, as presented in the simulation results, the steady state ripples of $i_L$ and $v_o$ increase with the decreasing of $\alpha$ and $\beta$, respectively. Moreover, the CCM-operating criterion is also order-dependent, and is mainly determined by the order $\alpha$ of the inductor. At the meantime, the line-to-output and duty cycle-to-output transfer functions of the DC-DC converters can be obtained from the analysis discussed in this paper, which both show a close relationship to the fractional orders. The generation condition of the asymptotically limit cycle behaviors, in the PI controlled fractional-order positive Luo converters, are also affected by the fractional orders. A smaller $\alpha$ or $\beta$, can help the converter to keep stability, and to increase the frequency of the asymptotically limit cycle $f_{acyc}$. In this paper, all the phenomena discussed above have been analyzed by the proposed method, of which the effectiveness has been verified by simulations and experiments.

In recent years, study results of the mathematical modeling of the passive components have shown that both the inductors and the capacitors are essentially of the fractional order. The modeling of the fractional-order converters has received widespread acceptance in engineering applications. Due to the great influence of the fractional orders to the properties of the power converters and their nonlinear dynamical behaviors, there is a big room for development and improvement in this research area. Based on the fact that the equivalent small parameter method and the principle of the harmonic balance are suitable for all DC-DC converters, the modified method proposed in this paper can be expanded to and cope with other fractional-order DC-DC converters. Moreover, with more precise equivalent circuit models of the electrolytic capacitors and the coil inductors, the benefits of using fractional-order modeling methodology can be further explored in the future.

## Supporting information

**S1 Appendix. The derivation of the solutions in Eqs (Eq (20a))–(20c).** The solutions of $\mathbf{x}_0$, $\mathbf{x}_1$, $\mathbf{x}_2$ in Eqs (Eq (20a))–(20c) can be obtained by the derivation.
(PDF)

**S2 Appendix. The approximation circuits of the fractional-order components.** The parameters in the approximation circuit of the fractional-order components are deduced.
(PDF)

**S1 File. *Matlab* c-script file to get the steady state variables using the proposed method.**
(M)

**S2 File. *Matlab* c-script file to obtain the steady state variables using the PECE-ABM method.**
(M)

**S3 File. *Matlab* Simulink model file to acquire the steady state variables using the modified Oustaloup's method.**
(MDL)

**S4 File. *Matlab* c-script file to get the bifurcation diagrams using the proposed method.**
(M)

**S5 File. *Matlab Simulink* model file to acquire the circuit simulation results.** This file is simulated using *PLECS Blockset*.
(MDL)

**S6 File. *PLECS* model file to obtain the circuit simulation results.** This file is simulated using *PLECS Standalone*.
(PLECS)

## Author Contributions

**Formal analysis:** Zirui Jia.

**Investigation:** Zirui Jia.

**Resources:** Zirui Jia.

**Software:** Zirui Jia.

**Supervision:** Chongxin Liu.

**Validation:** Chongxin Liu.

**Writing – original draft:** Zirui Jia.

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
