## [Decision Letter · Decision Letter 0]

19 May 2020

PONE-D-20-03914

A Modified Modeling and Dynamical Behavior Analysis Method for Fractional-order Positive Luo Converter

PLOS ONE

Dear Mrs Jia,

Thank you for submitting your manuscript to PLOS ONE. After careful consideration, we feel that it has merit but does not fully meet PLOS ONE’s publication criteria as it currently stands. Therefore, we invite you to submit a revised version of the manuscript that addresses the points raised during the review process.

ACADEMIC EDITOR: 

The reviewer raised concerns about this manuscript. The authors should revise the paper carefully. 

We would appreciate receiving your revised manuscript by Jul 03 2020 11:59PM. To enhance the reproducibility of your results, we recommend that if applicable you deposit your laboratory protocols in protocols.io, where a protocol can be assigned its own identifier (DOI) such that it can be cited independently in the future. For instructions see: http://journals.plos.org/plosone/s/submission-guidelines#loc-laboratory-protocols

We look forward to receiving your revised manuscript.

Kind regards,

Long Wang, Ph.D.

Academic Editor

PLOS ONE

"This work was supported in part by the National Nature Science Foundation of China under Grant No. 51877162."

"The author received no specific funding for this work."

Reviewers' comments:

Reviewer's Responses to Questions

**Comments to the Author**

1. Is the manuscript technically sound, and do the data support the conclusions?

Reviewer #1: Yes

2. Has the statistical analysis been performed appropriately and rigorously? 

Reviewer #1: Yes

3. Have the authors made all data underlying the findings in their manuscript fully available?

Reviewer #1: Yes

4. Is the manuscript presented in an intelligible fashion and written in standard English?

Reviewer #1: Yes

5. Review Comments to the Author

Reviewer #1: 1) The results about the conventional schemes and the proposed method should be more discussed.

2) It is important to have an experiment to verify the efficiency of the proposed method.

3) It should be included more references from 2019.

6. PLOS authors have the option to publish the peer review history of their article (what does this mean?). If published, this will include your full peer review and any attached files.

Reviewer #1: No

---

## [Author Response · Author response to Decision Letter 0]

2 Jul 2020

Reply to the academic editor:

Thank you very much for your requirements that greatly helped to improve our manuscript. At the same time, we should thank for your time and efforts.

We have studied your requirements carefully and have made correction which we hope meet with your approval. 

Response to requirement 1: Please ensure that your manuscript meets PLOS ONE's style requirements, including those for file naming. The PLOS ONE style templates can be found at http://www.plosone.org/attachments/PLOSOne_formatting_sample_main_body.pdf and http://www.plosone.org/attachments/PLOSOne_formatting_sample_title_authors_affiliations.pdf

Response: Terribly sorry for not meeting PLOS ONE's style requirements. We have carefully checked the format of the manuscript according to the PLOS ONE style templates. What’s more, the names of supplemental files in supporting information section are modified. These changes are marked in the revised manuscript. If you have any questions about the format of the manuscript, please do not hesitate to contact me.

Response to requirement 2: Thank you for stating the following in the Acknowledgments Section of your manuscript:

"This work was supported in part by the National Nature Science Foundation of China under Grant No. 51877162."

"The author received no specific funding for this work."

Response: We are very sorry for misunderstanding and not providing funding information in the Funding Statement section. The funding-related text in the manuscript is removed in the revised manuscript. And the Funding Information in the online submission form is modified. However, we cannot find Financial Disclosure in the Additional Information Section in Revised submission form. So, Funding Statement in the approval PDF still reads as follows:

"The author received no specific funding for this work."

We are terribly sorry for not modifying the Funding Statement. We hope this does not affect your judgment on our manuscript. At the same time, we hope the Funding Statement could be updated as follows:

"This work was supported in part by the National Nature Science Foundation of China under Grant No. 51877162, http://www.nsfc.gov.cn. The funders had no role in study design, data collection and analysis, decision to publish, or preparation of the manuscript. No additional external funding received for this study."

 Reply to reviewer #1

Thanks very much for your valuable comments and good suggestions that greatly helped to improve our manuscript. At the same time, we should thank for your time and efforts.

We have carefully considered your valuable comments and good suggestions. In the following, we are going to explain how your comments have been taken into account. 

Response to comment 1: The results about the conventional schemes and the proposed method should be more discussed.

Response: We are very sorry for not fully discussing the results about the conventional schemes and the proposed method. There are four methods we listed in the comparison section, the modified Oustaloup’s approximation, the Predictor-Corrector Adams-Bashforth-Moulton (PECE-ABM) type numerical method, the simplified equivalent small parameter (SESP) method and the proposed method in this paper. The advantages and disadvantages of these methods are listed in Table 1.

Table 1 Comparison of the four methods.

Method Advantages Disadvantages

Modified Oustaloup’s approximation method Without discretizing, this method is a precise engineering simulation for the transient responses analysis. 1. It is not an appropriate solution for the non-linear behaviors analysis due to the frequency domain approximation.

2. Complex fractional-order component approximation and circuit simulation should be used to obtain the steady state solutions.

PECE-ABM method It is a simple discrete model to obtain transient solutions 1. This method cannot track the dramatic changes in the transient solutions closely.

2. Large amounts of computation efforts are involved in the approach to get the steady-state variables.

SESP method 1. It does not need to use the fractional-order derivative definitions.

2. It is able to solve the steady state variables without circuit simulations or multiple iterations. 1. Its accuracy still needs improvement.

2. It does not apply to the situation in which the system state variable changes abruptly in one switching cycle.

3. The transient solutions of the fractional model are not mentioned.

Proposed method 1. It is a very accurate time-domain method especially in the situation when the system state variable changes abruptly in one switching cycle.

2. With the explicit Grünwald-Letnikov (GL) approximation, the proposed method can uncover the nonlinear fractional-order behaviors more comprehensively and realistically.

3. It does not need to use the fractional-order derivative definitions.

4. It is able to solve the solutions without circuit simulations or multiple iterations. 

Based on the above comparison, we modified the discussion of the results about the conventional schemes and the proposed method.

In the section 4 (Comparison and simulation of the conventional schemes and the proposed method), we have revise the results discussion in both of the two subsections. In subsection A (Comparison of the steady-state solutions), we have done more discussions about Fig 3, Fig 4 and Fig 6. The description before and after revisions are listed as follows:

1) Discussions about Fig 3, Fig 4

Before revision:

As shown in Fig 3 and Fig 4, the waveforms from these three schemes are consistent with each other, and the steady state ripples are influenced by the fractional orders. Specifically, the steady state ripples of iL and vo increase with the decreasing of α and β respectively. In general, the steady state ripples gotten by the proposed scheme are more closely resemble to the simulation results obtained by the modified Oustaloup's method, especially with smaller values of α and β. For the PECE-ABM method, the rangeabilities of both the DC components and the AC components are all undersized when α or β changes. This proves that the proposed method could track the dramatic changes more closely compared to the PECE-ABM method.

After revision:

As shown in Fig 3 and Fig 4, the waveforms from these three schemes are consistent with each other, and the steady state ripples are influenced by the fractional orders. Specifically, the steady state ripples of iL and vo increase with the decreasing of α and β respectively. In general, the steady state ripples gotten by the proposed scheme are more closely resemble to the simulation results obtained by the modified Oustaloup's method, especially with smaller values of α and β. For the PECE-ABM method, the rangeabilities of both the DC components and the AC components are all undersized when α or β changes. This proves that the proposed method can track the dramatic changes more closely compared to the PECE-ABM method. Reducing the step size can improve the accuracy of the PECE-ABM, but the number of iterative computations will be greatly increased at the same time. For the modified Oustaloup’s method, although it is more accurate, complex fractional-order component approximation and circuit simulation should be used to obtain the steady state solutions.

2) Discussions about Fig 6

Before revision:

In order to comprehensively compare the steady state solutions obtained by the SESP method, the Oustaloup’s approximation method and the proposed scheme, we consider the case in which (α, β, γ) = (0.9, 0.8, 0.95). Fig. 6 shows the comparison of these three schemes, which obviously shows that the steady state solutions obtained by the proposed method can be more closely resemble to the results gotten by the Oustaloup’s approximation method, compared to the SESP method, especially for vb. When the power switch turns off, the gap between the results gotten by the SESP method and the proposed scheme could be greater than 3%.

After revision:

In order to comprehensively compare the steady state solutions obtained by the SESP method, the Oustaloup’s approximation method and the proposed scheme, we consider the case in which (α, β, γ) = (0.9, 0.8, 0.95). Fig. 6 shows the comparison of these three schemes, which obviously shows that the steady state solutions obtained by the proposed method can be more closely resemble to the results gotten by the Oustaloup’s approximation method, compared to the SESP method, especially for vb. When the power switch turns off, the gap between the results gotten by the SESP method and the proposed scheme can be greater than 3%. Similar to the tolerance error index of ami，the tolerance error indexes of each state variable are calculated. It can be obtain that and . For vb, the lack of significant decrease in tolerance error index indicates that the number of iterations for SESP method is not enough. Thus, the corrections of DC value and main wave proposed in this paper are necessary.

In subsection B (Transient solutions comparison), further discussions about Fig 7 have done. The description before and after the modification is listed as follows:

3) Discussions about Fig 7

Before revision:

The comparison of the state variables versus n (n = (1000f)-1t), between the SESP method, the Oustaloup’s approximation method and the proposed scheme, is depicted in Fig 7. It can be seen that, the transient solutions of the proposed method are in good accordance with those from the Oustaloup’s approximation method. Because the explicit Grünwald-Letnikov (GL) approximation is used, the steady state solutions obtained by the proposed method are more accurate compared to the SESP method, especially for vb. The number of iterations is greatly reduced since the step size is not required to be smaller than the switching cycle.

After revision:

The comparison of the state variables versus n (n = (1000f)-1t), between the SESP method, the Oustaloup’s approximation method and the proposed scheme, is depicted in Fig 7. It can be seen that, the transient solutions of the proposed method are in good accordance with those from the Oustaloup’s approximation method. Based on the analysis of the previous subsection, using the proposed method, the numerical solution of each cycle is more accurate than SESP method. Because the explicit Grünwald-Letnikov (GL) approximation is used in the transient solution calculation, this error will accumulate with the increase of iterations, and be more apparent in the steady state. Therefore, the steady state solutions obtained by the proposed method are more accurate compared to the SESP method, especially for vb. When the power switch turns on, the gap between the SESP method and the proposed method is more obvious than that shown in Fig 6. The error between the maximum values of vb obtained by the two methods can exceed 1%. Moreover, the vb waveform obtained by the SESP method during the switching on conduction appears obvious distortion, which does not conform to the actual situation. Expect for the increased accuracy, the number of iterations is greatly reduced using proposed method, since the step size is not required to be smaller than the switching cycle.

In addition, more discussions about Fig 13 and 15 have done in section 6 (Dynamical behavior analysis of the PI controlled fractional-order positive Luo converter),. The description before and after the modification is listed as follows:

4) Discussions about Fig 15

Before revision:

The phase graph (Fig 15(a)) seems to be a trajectory of the periodic oscillation (solar to the limit cycle), while actually this oscillation is not perfectly periodic but is generating non-periodic items obviously. 

After revision:

The phase graph (Fig 15(a)) seems to be a trajectory of the periodic oscillation (solar to the limit cycle), while actually this oscillation is not perfectly periodic but is generating non-periodic items obviously. If the approximation of the fractional-order components in [13] is used, the phase graph will be a closed curve like Fig 13(a), which does not satisfy the well-known fact that the fractional-order systems cannot have any periodic solutions. 

Response to comment 2: It is important to have an experiment to verify the efficiency of the proposed method.

Response: We are terribly sorry for not providing experimental results due to a series of objective reasons, such as lack of experimental materials. We agree with your comment that an experiment to verify the efficiency of the proposed method is very important. Thus, a positive Luo converter prototype based on STB18N20 and MUR160 is built. And some techniques to reduce the interference of inductors and the parasitic parameters of the devices in the chain structure are introduced. In the revised paper, some experimental results are added in the section 7 (Circuit simulation and experimental results). Unfortunately, since the equivalent circuits of the fractional-order components are used in the experiment, the asymptotically period oscillation is not very obvious in the experimental results. The experimental results subsection in the revised manuscript is listed as follows: 

Experimental Results

In order to further verify the effectiveness of the proposed method and to realize the practical performance of fractional-order components, a positive Luo converter prototype is built. Fig 20 shows the photograph of the positive Luo converter. In this prototype, power switch and diode in Fig 1 choose STB18N20 and MUR160 respectively. To reduce the interference of inductors in the equivalent implementation circuit of fractional-order inductor, some shielded techniques, such as shielded power inductors are utilized.

Fig 20. Photograph of the prototype positive Luo converter.

Fig 21 shows the steady-state wave forms of iL and vo, which is obtained by experimental test with different fractional orders. Experimental measurements of these state variables are listed in Table 2. From Figs 21(a)-(d) and Table 2, it can be seen that the RMS and peak-peak value of iL and vo are basically consistent with the simulation analysis. Considering the influence of parasitic parameters and electromagnetic interference, the error between them can be ignored.

Fig 21. Experimental results under different fractional orders: (a) (α, β, γ) = (1, 1, 1); (b) (α, β, γ) = (0.9, 1, 1); (c) (α, β, γ) = (1, 0.9, 1); (b) (α, β, γ) = (0.9, 0.9, 1).

Table 2. Experimental results under different fractional orders.

(α, β, γ) iL_RMS ΔiL_pp vo_RMS Δvo_pp

(1, 1, 1) 1.112 0.062 29.42 1.83

(0.9, 1, 1) 1.131 0.098 29.23 1.87

(1, 0.9, 1) 1.096 0.196 29.16 4.79

(0.9, 0.9, 1) 1.097 0.198 28.95 4.84

Response to comment 3: It should be included more references from 2019.

Response: We should apologize for not using the latest literature about the application of fractional-order. After fully considering your comments, we replaced the Ref. [1, 2, 3, 4, 20, 21, 24, 28] with eight references published in 2019 and 2020. These eight references in the revised paper are listed as follows:

1. Xiong R, Tian J, Shen W, et al. A Novel Fractional Order Model for State of Charge Estimation in Lithium Ion Batteries. IEEE Transactions on Vehicular Technology. 2019; 68(5):4130-4139.

2. Kang S, Wu H, Yang X, et al. Fractional-order robust model reference adaptive control of piezo-actuated active vibration isolation systems using output feedback and multi-objective optimization algorithm. Journal of Vibration and Control. 2019; 26(2):107754631987526.

3. Khan A, Gómez-Aguilar, J.F, Saeed Khan T, et al. Stability analysis and numerical solutions of fractional order HIV/AIDS model. Chaos Solitons & Fractals. 2019; 122:119-128.

4. Wang H, Gu Y, Yu Y. Numerical solution of fractional-order time-varying delayed differential systems using Lagrange interpolation. Nonlinear Dynamics. 2019; 95:809-822.

20. Ali Yüce, Tan N. Electronic realisation technique for fractional order integrators. The Journal of Engineering. 2020; 2020(5):157-167.

21. Buscarino A, Caponetto R, Graziani S, et al. Realization of fractional order circuits by a Constant Phase Element. European Journal of Control. 2019; 54:64-72.

24. Mahata S, Saha S K, Kar R, et al. Approximation of fractional-order low-pass filter. IET Signal Processing. 2019; 13(1):112-124.

28. Aldo Jonathan Muoz-Vázquez, Sanchez-Orta A, Parra-Vega V. A general result on non-existence of finite-time stable equilibria in fractional-order systems. Journal of the Franklin Institute. 2019; 356(1):268-275.

We tried our best to improve the manuscript and made some changes in the manuscript. These changes will not influence the content and framework of the paper. And here we did not list the changes but marked in revised paper.

---

## [Editor Report · Decision Letter 1]

22 Jul 2020

A Modified Modeling and Dynamical Behavior Analysis Method for Fractional-order Positive Luo Converter

PONE-D-20-03914R1

Dear Dr. Jia,

We’re pleased to inform you that your manuscript has been judged scientifically suitable for publication and will be formally accepted for publication once it meets all outstanding technical requirements.

Kind regards,

Long Wang, Ph.D.

Academic Editor

PLOS ONE
---

## [Editor Report · Acceptance letter]

27 Jul 2020

PONE-D-20-03914R1 

A Modified Modeling and Dynamical Behavior Analysis Method for Fractional-order Positive Luo Converter 

Dear Dr. Jia:

I'm pleased to inform you that your manuscript has been deemed suitable for publication in PLOS ONE. Congratulations! Your manuscript is now with our production department. 

Kind regards, 

on behalf of

Dr. Long Wang 

Academic Editor

PLOS ONE